# Robust Regression via Hard Thresholding

**Kush Bhatia**[†]**, Prateek Jain**[†]**, and Purushottam Kar**[‡*]
[†]Microsoft Research, India
[‡]Indian Institute of Technology Kanpur, India
{t-kushb,prajain}@microsoft.com, purushot@cse.iitk.ac.in

## Abstract

We study the problem of Robust Least Squares Regression (RLSR) where several response variables can be adversarially corrupted. More specifically, for a data matrix $X \in \mathbb{R}^{p \times n}$ and an underlying model $\mathbf{w}^*$, the response vector is generated as $\mathbf{y} = X^T \mathbf{w}^* + \mathbf{b}$ where $\mathbf{b} \in \mathbb{R}^n$ is the corruption vector supported over at most $C \cdot n$ coordinates. Existing exact recovery results for RLSR focus solely on $L_1$-penalty based convex formulations and impose relatively strict model assumptions such as requiring the corruptions $\mathbf{b}$ to be selected independently of $X$.

In this work, we study a simple hard-thresholding algorithm called TORRENT which, under mild conditions on $X$, can recover $\mathbf{w}^*$ exactly even if $\mathbf{b}$ corrupts the response variables in an *adversarial* manner, i.e. both the support and entries of $\mathbf{b}$ are selected adversarially after observing $X$ and $\mathbf{w}^*$. Our results hold under *deterministic* assumptions which are satisfied if $X$ is sampled from any sub-Gaussian distribution. Finally unlike existing results that apply only to a fixed $\mathbf{w}^*$, generated independently of $X$, our results are *universal* and hold for any $\mathbf{w}^* \in \mathbb{R}^p$.

Next, we propose gradient descent-based extensions of TORRENT that can scale efficiently to large scale problems, such as high dimensional sparse recovery. and prove similar recovery guarantees for these extensions. Empirically we find TORRENT, and more so its extensions, offering significantly faster recovery than the state-of-the-art $L_1$ solvers. For instance, even on moderate-sized datasets (with $p = 50K$) with around $40\%$ corrupted responses, a variant of our proposed method called TORRENT-HYB is more than $20\times$ faster than the best $L_1$ solver.

> *"If among these errors are some which appear too large to be admissible, then those equations which produced these errors will be rejected, as coming from too faulty experiments, and the unknowns will be determined by means of the other equations, which will then give much smaller errors."*
>
> A. M. Legendre, *On the Method of Least Squares.* 1805.

## 1 Introduction

Robust Least Squares Regression (RLSR) addresses the problem of learning a reliable set of regression coefficients in the presence of several arbitrary corruptions in the *response* vector. Owing to the wide-applicability of regression, RLSR features as a critical component of several important real-world applications in a variety of domains such as signal processing [1], economics [2], computer vision [3, 4], and astronomy [2].

Given a data matrix $X = [\mathbf{x}_1, \ldots, \mathbf{x}_n]$ with $n$ data points in $\mathbb{R}^p$ and the corresponding response vector $\mathbf{y} \in \mathbb{R}^n$, the goal of RLSR is to learn a $\hat{\mathbf{w}}$ such that,

$$(\hat{\mathbf{w}}, \hat{S}) = \underset{\substack{\mathbf{w} \in \mathbb{R}^p \\ S \subset [n]: |S| \geq (1-\beta) \cdot n}}{\arg\min} \sum_{i \in S} (y_i - \mathbf{x}_i^T \mathbf{w})^2, \tag{1}$$

That is, we wish to simultaneously determine the set of corruption free points $\hat{S}$ and also estimate the best model parameters over the set of clean points. However, the optimization problem given above is non-convex (jointly in $\mathbf{w}$ and $S$) in general and might not directly admit efficient solutions. Indeed there exist reformulations of this problem that are known to be NP-hard to optimize [1].

To address this problem, most existing methods with provable guarantees assume that the observations are obtained from some generative model. A commonly adopted model is the following

$$\mathbf{y} = X^T \mathbf{w}^* + \mathbf{b}, \tag{2}$$

where $\mathbf{w}^* \in \mathbb{R}^p$ is the *true* model vector that we wish to estimate and $\mathbf{b} \in \mathbb{R}^n$ is the corruption vector that can have arbitrary values. A common assumption is that the corruption vector is *sparsely supported* i.e. $\|\mathbf{b}\|_0 \leq \alpha \cdot n$ for some $\alpha > 0$.

Recently, [4] and [5] obtained a surprising result which shows that one can recover $\mathbf{w}^*$ *exactly* even when $\alpha \lesssim 1$, i.e., when almost all the points are corrupted, by solving an $L_1$-penalty based convex optimization problem: $\min_{\mathbf{w},\mathbf{b}} \|\mathbf{w}\|_1 + \lambda \|\mathbf{b}\|_1$, s.t., $X^\top \mathbf{w} + \mathbf{b} = \mathbf{y}$. However, these results require the corruption vector $\mathbf{b}$ to be selected oblivious of $X$ and $\mathbf{w}^*$. Moreover, the results impose severe restrictions on the data distribution, requiring that the data be either sampled from an isotropic Gaussian ensemble [4], or row-sampled from an incoherent orthogonal matrix [5]. Finally, these results hold only for a fixed $\mathbf{w}^*$ and are not universal in general.

In contrast, [6] studied RLSR with less stringent assumptions, allowing arbitrary corruptions in response variables as well as in the data matrix $X$, and proposed a trimmed inner product based algorithm for the problem. However, their recovery guarantees are significantly weaker. Firstly, they are able to recover $\mathbf{w}^*$ only upto an additive error $\alpha\sqrt{p}$ (or $\alpha\sqrt{s}$ if $\mathbf{w}^*$ is $s$-sparse). Hence, they require $\alpha \leq 1/\sqrt{p}$ just to claim a non-trivial bound. Note that this amounts to being able to tolerate only a vanishing fraction of corruptions. More importantly, even with $n \to \infty$ and extremely small $\alpha$ they are unable to guarantee exact recovery of $\mathbf{w}^*$. A similar result was obtained by [7], albeit using a sub-sampling based algorithm with stronger assumptions on $\mathbf{b}$.

In this paper, we focus on a simple and natural thresholding based algorithm for RLSR. At a high level, at each step $t$, our algorithm alternately estimates an *active set* $S_t$ of "clean" points and then updates the model to obtain $\mathbf{w}^{t+1}$ by minimizing the least squares error on the active set. This intuitive algorithm seems to embody a long standing heuristic first proposed by Legendre [8] over two centuries ago (see introductory quotation in this paper) that has been adopted in later literature [9, 10] as well. However, to the best of our knowledge, this technique has never been rigorously analyzed before in non-asymptotic settings, despite its appealing simplicity.

**Our Contributions**: The main contribution of this paper is an exact recovery guarantee for the thresholding algorithm mentioned above that we refer to as TORRENT-FC (see Algorithm 1). We provide our guarantees in the model given in 2 where the corruptions $\mathbf{b}$ are selected *adversarially* but restricted to have at most $\alpha \cdot n$ non-zero entries where $\alpha$ is a global constant dependent only on $X$[1]. Under *deterministic* conditions on $X$, namely the subset strong convexity (SSC) and smoothness (SSS) properties (see Definition 1), we guarantee that TORRENT-FC converges at a *geometric* rate and recovers $\mathbf{w}^*$ exactly. We further show that these properties (SSC and SSS) are satisfied w.h.p. if a) the data $X$ is sampled from a sub-Gaussian distribution and, b) $n \geq p \log p$.

We would like to stress three key advantages of our result over the results of [4, 5]: a) we allow $\mathbf{b}$ to be adversarial, i.e., both support and values of $\mathbf{b}$ to be selected adversarially based on $X$ and $\mathbf{w}^*$, b) we make assumptions on data that are natural, as well as significantly less restrictive than what existing methods make, and c) our analysis admits universal guarantees, i.e., holds for *any* $\mathbf{w}^*$.

We would also like to stress that while hard-thresholding based methods have been studied rigorously for the sparse-recovery problem [11, 12], hard-thresholding has not been studied formally for the robust regression problem. [13] study soft-thresholding approaches to the robust regression problem but without any formal guarantees. Moreover, the two problems are completely different and hence techniques from sparse-recovery analysis do not extend to robust regression.

Despite its simplicity, TORRENT-FC does not scale very well to datasets with large $p$ as it solves least squares problems at each iteration. We address this issue by designing a gradient descent based algorithm (TORRENT-GD), and a hybrid algorithm (TORRENT-Hyb), both of which enjoy a geometric rate of convergence and can recover $\mathbf{w}^*$ under the model assumptions mentioned above. We also propose extensions of TORRENT for the RLSR problem in the sparse regression setting where $p \gg n$ but $\|\mathbf{w}^*\|_0 = s^* \ll p$. Our algorithm TORRENT-HD is based on TORRENT-FC but uses the Iterative Hard Thresholding (IHT) algorithm, a popular algorithm for sparse regression. As before, we show that TORRENT-HD also converges geometrically to $\mathbf{w}^*$ if a) the corruption index $\alpha$ is less than some constant $C$, b) $X$ is sampled from a sub-Gaussian distribution and, c) $n \geq s^* \log p$.

Finally, we experimentally evaluate existing $L_1$-based algorithms and our hard thresholding-based algorithms. The results demonstrate that our proposed algorithms (TORRENT-(FC/GD/HYB)) can be significantly faster than the best $L_1$ solvers, exhibit better recovery properties, as well as be more robust to dense white noise. For instance, on a problem with $50K$ dimensions and $40\%$ corruption, TORRENT-HYB was found to be $20\times$ faster than $L_1$ solvers, as well as achieve lower error rates.

## 2 Problem Formulation

Given a set of data points $X = [\mathbf{x}_1, \mathbf{x}_2, \ldots, \mathbf{x}_n]$, where $\mathbf{x}_i \in \mathbb{R}^p$ and the corresponding response vector $\mathbf{y} \in \mathbb{R}^n$, the goal is to recover a parameter vector $\mathbf{w}^*$ which solves the RLSR problem (1). We assume that the response vector $\mathbf{y}$ is generated using the following model:

$$\mathbf{y} = \mathbf{y}^* + \mathbf{b} + \boldsymbol{\varepsilon}, \text{ where } \mathbf{y}^* = X^\top \mathbf{w}^*.$$

Hence, in the above model, (1) reduces to estimating $\mathbf{w}^*$. We allow the model $\mathbf{w}^*$ representing the regressor, to be chosen in an adaptive manner *after* the data features have been generated.

The above model allows two kinds of perturbations to $y_i$ – dense but bounded noise $\varepsilon_i$ (e.g. white noise $\varepsilon_i \sim \mathcal{N}(0, \sigma^2), \sigma \geq 0$), as well as potentially unbounded corruptions $b_i$ – to be introduced by an adversary. The only requirement we enforce is that the gross corruptions be sparse. $\boldsymbol{\varepsilon}$ shall represent the dense noise vector, for example $\boldsymbol{\varepsilon} \sim \mathcal{N}(\mathbf{0}, \sigma^2 \cdot I_{n \times n})$, and $\mathbf{b}$, the corruption vector such that $\|\mathbf{b}\|_0 \leq \alpha \cdot n$ for some *corruption index* $\alpha > 0$. We shall use the notation $S_* = \overline{\text{supp}(\mathbf{b})} \subseteq [n]$ to denote the set of "clean" points, i.e. points that have not faced unbounded corruptions. We consider adaptive adversaries that are able to view the generated data points $\mathbf{x}_i$, as well as the clean responses $y_i^*$ and dense noise values $\varepsilon_i$ before deciding which locations to corrupt and by what amount.

We denote the unit sphere in $p$ dimensions using $S^{p-1}$. For any $\gamma \in (0, 1]$, we let $\mathcal{S}_\gamma = \{S \subset [n] : |S| = \gamma \cdot n\}$ denote the set of all subsets of size $\gamma \cdot n$. For any set $S$, we let $X_S := [\mathbf{x}_i]_{i \in S} \in \mathbb{R}^{p \times |S|}$ denote the matrix whose columns are composed of points in that set. Also, for any vector $\mathbf{v} \in \mathbb{R}^n$ we use the notation $\mathbf{v}_S$ to denote the $|S|$-dimensional vector consisting of those components that are in $S$. We use $\lambda_{\min}(X)$ and $\lambda_{\max}(X)$ to denote, respectively, the smallest and largest eigenvalues of a square symmetric matrix $X$. We now introduce two properties, namely, *Subset Strong Convexity* and *Subset Strong Smoothness*, which are key to our analyses.

**Definition 1** (SSC and SSS Properties). *A matrix $X \in \mathbb{R}^{p \times n}$ satisfies the* Subset Strong Convexity Property *(resp.* Subset Strong Smoothness Property*) at level $\gamma$ with strong convexity constant $\lambda_\gamma$ (resp. strong smoothness constant $\Lambda_\gamma$) if the following holds:*

$$\lambda_\gamma \leq \min_{S \in \mathcal{S}_\gamma} \lambda_{\min}(X_S X_S^\top) \leq \max_{S \in \mathcal{S}_\gamma} \lambda_{\max}(X_S X_S^\top) \leq \Lambda_\gamma.$$

*Remark* 1. We note that the uniformity enforced in the definitions of the SSC and SSS properties is not for the sake of convenience but rather a necessity. Indeed, a uniform bound is required in face of an adversary which can perform corruptions *after* data and response variables have been generated, and choose to corrupt precisely that set of points where the SSC and SSS parameters are the worst.

## 3 TORRENT: Thresholding Operator-based Robust Regression Method

We now present TORRENT, a Thresholding Operator-based Robust RegrEssioN meThod for performing robust regression at scale. Key to our algorithms is the *Hard Thresholding Operator* which we define below.

**Algorithm 1** TORRENT: Thresholding Operator-based Robust RegrEssioN meThod

**Input:** Training data $\{\mathbf{x}_i, y_i\}, i = 1 \ldots n$, step length $\eta$, thresholding parameter $\beta$, tolerance $\epsilon$
1: $\mathbf{w}^0 \leftarrow \mathbf{0}, S_0 = [n], t \leftarrow 0, \mathbf{r}^0 \leftarrow \mathbf{y}$
2: **while** $\left\| \mathbf{r}^t_{S_t} \right\|_2 > \epsilon$ **do**
3: $\quad \mathbf{w}^{t+1} \leftarrow$ UPDATE$(\mathbf{w}^t, S_t, \eta, \mathbf{r}^t, S_{t-1})$
4: $\quad r_i^{t+1} \leftarrow \left( y_i - \left\langle \mathbf{w}^{t+1}, \mathbf{x}_i \right\rangle \right)$
5: $\quad S_{t+1} \leftarrow$ HT$(\mathbf{r}^{t+1}, (1 - \beta)n)$
6: $\quad t \leftarrow t + 1$
7: **end while**
8: **return** $\mathbf{w}^t$

---

**Algorithm 2** UPDATE TORRENT-FC

**Input:** Current model $\mathbf{w}$, current active set $S$
1: **return** $\arg\min_{\mathbf{w}} \sum_{i \in S} (y_i - \langle \mathbf{w}, \mathbf{x}_i \rangle)^2$

---

**Algorithm 3** UPDATE TORRENT-GD

**Input:** Current model $\mathbf{w}$, current active set $S$, step size $\eta$
1: $\mathbf{g} \leftarrow X_S(X_S^\top \mathbf{w} - \mathbf{y}_S)$
2: **return** $\mathbf{w} - \eta \cdot \mathbf{g}$

---

**Algorithm 4** UPDATE TORRENT-HYB

**Input:** Current model $\mathbf{w}$, current active set $S$, step size $\eta$, current residuals $\mathbf{r}$, previous active set $S'$
1: // Use the GD update if the active set $S$ is changing a lot
2: **if** $|S \backslash S'| > \Delta$ **then**
3: $\quad \mathbf{w}' \leftarrow$ UPDATE-GD$(\mathbf{w}, S, \eta, \mathbf{r}, S')$
4: **else**
5: // If stable, use the FC update
6: $\quad \mathbf{w}' \leftarrow$ UPDATE-FC$(\mathbf{w}, S)$
7: **end if**
8: **return** $\mathbf{w}'$

---

**Definition 2** (Hard Thresholding Operator). *For any vector $\mathbf{v} \in \mathbb{R}^n$, let $\sigma_{\mathbf{v}} \in S_n$ be the permutation that orders elements of $\mathbf{v}$ in ascending order of their magnitudes i.e. $\left| \mathbf{v}_{\sigma_{\mathbf{v}}(1)} \right| \leq \left| \mathbf{v}_{\sigma_{\mathbf{v}}(2)} \right| \leq \ldots \leq \left| \mathbf{v}_{\sigma_{\mathbf{v}}(n)} \right|$. Then for any $k \leq n$, we define the hard thresholding operator as*

$$\text{HT}(\mathbf{v}; k) = \left\{ i \in [n] : \sigma_{\mathbf{v}}^{-1}(i) \leq k \right\}$$

Using this operator, we present our algorithm TORRENT (Algorithm 1) for robust regression. TORRENT follows a most natural iterative strategy of, alternately, estimating an *active set* of points which have the least residual error on the current regressor, and then updating the regressor to provide a better fit on this active set. We offer three variants of our algorithm, based on how aggressively the algorithm tries to fit the regressor to the current active set.

We first propose a fully corrective algorithm TORRENT-FC (Algorithm 2) that performs a fully corrective least squares regression step in an effort to minimize the regression error on the active set. This algorithm makes significant progress in each step, but at a cost of more expensive updates. To address this, we then propose a milder, gradient descent-based variant TORRENT-GD (Algorithm 3) that performs a much cheaper update of taking a single step in the direction of the gradient of the objective function on the active set. This reduces the regression error on the active set but does not minimize it. This turns out to be beneficial in situations where dense noise is present along with sparse corruptions since it prevents the algorithm from overfitting to the current active set.

Both the algorithms proposed above have their pros and cons – the FC algorithm provides significant improvements with each step, but is expensive to execute whereas the GD variant, although efficient in executing each step, offers slower progress. To get the best of both these algorithms, we propose a third, hybrid variant TORRENT-HYB (Algorithm 4) that adaptively selects either the FC or the GD update depending on whether the active set is stable across iterations or not.

In the next section we show that this hard thresholding-based strategy offers a linear convergence rate for the algorithm in all its three variations. We shall also demonstrate the applicability of this technique to high dimensional sparse recovery settings in a subsequent section.

## 4 Convergence Guarantees

For the sake of ease of exposition, we will first present our convergence analyses for cases where dense noise is not present i.e. $\mathbf{y} = X^\top \mathbf{w}^* + \mathbf{b}$ and will handle cases with dense noise *and* sparse corruptions later. We first analyze the fully corrective TORRENT-FC algorithm. The convergence proof in this case relies on the optimality of the two steps carried out by the algorithm, the fully corrective step that selects the best regressor on the active set, and the hard thresholding step that discovers a new active set by selecting points with the least residual error on the current regressor.

**Theorem 3.** *Let $X = [\mathbf{x}_1, \ldots, \mathbf{x}_n] \in \mathbb{R}^{p \times n}$ be the given data matrix and $\mathbf{y} = X^T \mathbf{w}^* + \mathbf{b}$ be the corrupted output with $\|\mathbf{b}\|_0 \leq \alpha \cdot n$. Let Algorithm 2 be executed on this data with the thresholding parameter set to $\beta \geq \alpha$. Let $\Sigma_0$ be an invertible matrix such that $\widetilde{X} = \Sigma_0^{-1/2} X$ satisfies the SSC and SSS properties at level $\gamma$ with constants $\lambda_\gamma$ and $\Lambda_\gamma$ respectively (see Definition 1). If the data satisfies $\frac{(1+\sqrt{2})\Lambda_\beta}{\lambda_{1-\beta}} < 1$, then after $t = \mathcal{O}\left(\log\left(\frac{1}{\sqrt{n}}\frac{\|\mathbf{b}\|_2}{\epsilon}\right)\right)$ iterations, Algorithm 2 obtains an $\epsilon$-accurate solution $\mathbf{w}^t$ i.e. $\|\mathbf{w}^t - \mathbf{w}^*\|_2 \leq \epsilon$.*

*Proof (Sketch).* Let $\mathbf{r}^t = \mathbf{y} - X^\top \mathbf{w}^t$ be the vector of residuals at time $t$ and $C_t = X_{S_t} X_{S_t}^\top$. Also let $S_* = \overline{\text{supp}(\mathbf{b})}$ be the set of uncorrupted points. The fully corrective step ensures that

$$\mathbf{w}^{t+1} = C_t^{-1} X_{S_t} \mathbf{y}_{S_t} = C_t^{-1} X_{S_t} \left(X_{S_t}^\top \mathbf{w}^* + \mathbf{b}_{S_t}\right) = \mathbf{w}^* + C_t^{-1} X_{S_t} \mathbf{b}_{S_t},$$

whereas the hard thresholding step ensures that $\left\|\mathbf{r}_{S_{t+1}}^{t+1}\right\|_2^2 \leq \left\|\mathbf{r}_{S_*}^{t+1}\right\|_2^2$. Combining the two gives us

$$\|\mathbf{b}_{S_{t+1}}\|_2^2 \leq \left\|X_{S_* \setminus S_{t+1}}^\top C_t^{-1} X_{S_t} \mathbf{b}_{S_t}\right\|_2^2 + 2 \cdot \mathbf{b}_{S_{t+1}}^\top X_{S_{t+1}}^\top C_t^{-1} X_{S_t} \mathbf{b}_{S_t}$$

$$\overset{\zeta_1}{=} \left\|\widetilde{X}_{S_* \setminus S_{t+1}}^\top \left(\widetilde{X}_{S_t} \widetilde{X}_{S_t}^T\right)^{-1} \widetilde{X}_{S_t} \mathbf{b}_{S_t}\right\|_2^2 + 2 \cdot \mathbf{b}_{S_{t+1}}^\top \widetilde{X}_{S_{t+1}}^\top \left(\widetilde{X}_{S_t} \widetilde{X}_{S_t}^T\right)^{-1} \widetilde{X}_{S_t} \mathbf{b}_{S_t}$$

$$\overset{\zeta_2}{\leq} \frac{\Lambda_\beta^2}{\lambda_{1-\beta}^2} \cdot \|\mathbf{b}_{S_t}\|_2^2 + 2 \cdot \frac{\Lambda_\beta}{\lambda_{1-\beta}} \cdot \|\mathbf{b}_{S_t}\|_2 \|\mathbf{b}_{S_{t+1}}\|_2,$$

where $\zeta_1$ follows from setting $\widetilde{X} = \Sigma_0^{-1/2} X$ and $X_S^\top C_t^{-1} X_{S'} = \widetilde{X}_S^\top (\widetilde{X}_{S_t} \widetilde{X}_{S_t}^\top)^{-1} \widetilde{X}_{S'}$ and $\zeta_2$ follows from the SSC and SSS properties, $\|\mathbf{b}_{S_t}\|_0 \leq \|\mathbf{b}\|_0 \leq \beta \cdot n$ and $|S_* \setminus S_{t+1}| \leq \beta \cdot n$. Solving the quadratic equation and performing other manipulations gives us the claimed result. $\square$

Theorem 3 relies on a deterministic (*fixed design*) assumption, specifically $\frac{(1+\sqrt{2})\Lambda_\beta}{\lambda_{1-\beta}} < 1$ in order to guarantee convergence. We can show that a large class of random designs, including Gaussian and sub-Gaussian designs actually satisfy this requirement. That is to say, data generated from these distributions satisfy the SSC and SSS conditions such that $\frac{(1+\sqrt{2})\Lambda_\beta}{\lambda_{1-\beta}} < 1$ with high probability. Theorem 4 explicates this for the class of Gaussian designs.

**Theorem 4.** *Let $X = [\mathbf{x}_1, \ldots, \mathbf{x}_n] \in \mathbb{R}^{p \times n}$ be the given data matrix with each $\mathbf{x}_i \sim \mathcal{N}(\mathbf{0}, \Sigma)$. Let $\mathbf{y} = X^\top \mathbf{w}^* + \mathbf{b}$ and $\|\mathbf{b}\|_0 \leq \alpha \cdot n$. Also, let $\alpha \leq \beta < \frac{1}{65}$ and $n \geq \Omega\left(p + \log \frac{1}{\delta}\right)$. Then, with probability at least $1 - \delta$, the data satisfies $\frac{(1+\sqrt{2})\Lambda_\beta}{\lambda_{1-\beta}} < \frac{9}{10}$. More specifically, after $T \geq 10 \log\left(\frac{1}{\sqrt{n}}\frac{\|\mathbf{b}\|_2}{\epsilon}\right)$ iterations of Algorithm 1 with the thresholding parameter set to $\beta$, we have $\left\|\mathbf{w}^T - \mathbf{w}^*\right\| \leq \epsilon$.*

*Remark* 2. Note that Theorem 4 provides rates that are independent of the condition number $\frac{\lambda_{\max}(\Sigma)}{\lambda_{\min}(\Sigma)}$ of the distribution. We also note that results similar to Theorem 4 can be proven for the larger class of sub-Gaussian distributions. We refer the reader to Section G for the same.

*Remark* 3. We remind the reader that our analyses can readily accommodate dense noise in addition to sparse unbounded corruptions. We direct the reader to Appendix A which presents convergence proofs for our algorithms in these settings.

*Remark* 4. We would like to point out that the design requirements made by our analyses are very mild when compared to existing literature. Indeed, the work of [4] assumes the *Bouquet Model* where distributions are restricted to be isotropic Gaussians whereas the work of [5] assumes a more stringent model of sub-orthonormal matrices, something that even Gaussian designs do not satisfy. Our analyses, on the other hand, hold for the general class of sub-Gaussian distributions.

We now analyze the TORRENT-GD algorithm which performs cheaper, gradient-style updates on the active set. We will show that this method nevertheless enjoys a linear rate of convergence.

**Theorem 5.** *Let the data settings be as stated in Theorem 3 and let Algorithm 3 be executed on this data with the thresholding parameter set to $\beta \geq \alpha$ and the step length set to $\eta = \frac{1}{\Lambda_{1-\beta}}$. If the data*

satisfies $\max\left\{\eta\sqrt{\Lambda_\beta}, 1-\eta\lambda_{1-\beta}\right\} \le \frac{1}{4}$, then after $t = \mathcal{O}\left(\log\left(\frac{\|b\|_2}{\sqrt{n}}\frac{1}{\epsilon}\right)\right)$ iterations, Algorithm 1 obtains an $\epsilon$-accurate solution $\mathbf{w}^t$ i.e. $\|\mathbf{w}^t - \mathbf{w}^*\|_2 \le \epsilon$.

Similar to TORRENT-FC, the assumptions made by the TORRENT-GD algorithm are also satisfied by the class of sub-Gaussian distributions. The proof of Theorem 5, given in Appendix D, details these arguments. Given the convergence analyses for TORRENT-FC and GD, we now move on to provide a convergence analysis for the hybrid TORRENT-HYB algorithm which interleaves FC and GD steps. Since the exact interleaving adopted by the algorithm depends on the data, and not known in advance, this poses a problem. We address this problem by giving below a uniform convergence guarantee, one that applies to *every interleaving* of the FC and GD update steps.

**Theorem 6.** *Suppose Algorithm 4 is executed on data that allows Algorithms 2 and 3 a convergence rate of $\eta_{\mathrm{FC}}$ and $\eta_{\mathrm{GD}}$ respectively. Suppose we have $2\cdot\eta_{\mathrm{FC}}\cdot\eta_{\mathrm{GD}} < 1$. Then for* any *interleavings of the FC and GD steps that the policy may enforce, after $t = \mathcal{O}\left(\log\left(\frac{1}{\sqrt{n}}\frac{\|\mathbf{b}\|_2}{\epsilon}\right)\right)$ iterations, Algorithm 4 ensures an $\epsilon$-optimal solution i.e. $\|\mathbf{w}^t - \mathbf{w}^*\| \le \epsilon$.*

We point out to the reader that the assumption made by Theorem 6 i.e. $2 \cdot \eta_{\mathrm{FC}} \cdot \eta_{\mathrm{GD}} < 1$ is readily satisfied by random sub-Gaussian designs, albeit at the cost of reducing the noise tolerance limit. As we shall see, TORRENT-HYB offers attractive convergence properties, merging the fast convergence rates of the FC step, as well as the speed and protection against overfitting provided by the GD step.

## 5 High-dimensional Robust Regression

In this section, we extend our approach to the robust high-dimensional sparse recovery setting. As before, we assume that the response vector $\mathbf{y}$ is obtained as: $\mathbf{y} = X^\top \mathbf{w}^* + \mathbf{b}$, where $\|\mathbf{b}\|_0 \le \alpha \cdot n$. However, this time, we also assume that $\mathbf{w}^*$ is $s^*$-sparse i.e. $\|\mathbf{w}^*\|_0 \le s^*$. As before, we shall neglect white/dense noise for the sake of simplicity. We reiterate that it is not possible to use existing results from sparse recovery (such as [11, 12]) directly to solve this problem.

Our objective would be to recover a *sparse* model $\hat{\mathbf{w}}$ so that $\|\hat{\mathbf{w}} - \mathbf{w}^*\|_2 \le \epsilon$. The challenge here is to forgo a sample complexity of $n \gtrsim p$ and instead, perform recovery with $n \sim s^* \log p$ samples alone. For this setting, we modify the FC update step of TORRENT-FC method to the following:

$$\mathbf{w}^{t+1} \leftarrow \inf_{\|\mathbf{w}\|_0 \le s} \sum_{i \in S_t} \left(y_i - \langle \mathbf{w}, \mathbf{x}_i \rangle\right)^2, \tag{3}$$

for some *target* sparsity level $s \ll p$. We refer to this modified algorithm as TORRENT-HD. Assuming $X$ satisfies the RSC/RSS properties (defined below), (3) can be solved efficiently using results from sparse recovery (for example the IHT algorithm [11, 14] analyzed in [12]).

**Definition 7** (RSC and RSS Properties). *A matrix $X \in \mathbb{R}^{p \times n}$ will be said to satisfy the Restricted Strong Convexity Property (resp. Restricted Strong Smoothness Property) at level $s = s_1 + s_2$ with strong convexity constant $\alpha_{s_1+s_2}$ (resp. strong smoothness constant $L_{s_1+s_2}$) if the following holds for all $\|\mathbf{w}_1\|_0 \le s_1$ and $\|\mathbf{w}_2\|_0 \le s_2$:*

$$\alpha_s \|\mathbf{w}_1 - \mathbf{w}_2\|_2^2 \le \left\|X^\top(\mathbf{w}_1 - \mathbf{w}_2)\right\|_2^2 \le L_s \|\mathbf{w}_1 - \mathbf{w}_2\|_2^2$$

For our results, we shall require the subset versions of both these properties.

**Definition 8** (SRSC and SRSS Properties). *A matrix $X \in \mathbb{R}^{p \times n}$ will be said to satisfy the Subset Restricted Strong Convexity (resp. Subset Restricted Strong Smoothness) Property at level $(\gamma, s)$ with strong convexity constant $\alpha_{(\gamma,s)}$ (resp. strong smoothness constant $L_{(\gamma,s)}$) if for all subsets $S \in \mathcal{S}_\gamma$, the matrix $X_S$ satisfies the RSC (resp. RSS) property at level $s$ with constant $\alpha_s$ (resp. $L_s$).*

We now state the convergence result for the TORRENT-HD algorithm.

**Theorem 9.** *Let $X \in \mathbb{R}^{p \times n}$ be the given data matrix and $\mathbf{y} = X^T \mathbf{w}^* + \mathbf{b}$ be the corrupted output with $\|\mathbf{w}^*\|_0 \le s^*$ and $\|\mathbf{b}\|_0 \le \alpha \cdot n$. Let $\Sigma_0$ be an invertible matrix such that $\Sigma_0^{-1/2} X$ satisfies the SRSC and SRSS properties at level $(\gamma, 2s+s^*)$ with constants $\alpha_{(\gamma,2s+s^*)}$ and $L_{(\gamma,2s+s^*)}$ respectively (see Definition 8). Let Algorithm 2 be executed on this data with the TORRENT-HD update, thresholding parameter set to $\beta \ge \alpha$, and $s \ge 32\left(\frac{L_{(1-\beta,2s+s^*)}}{\alpha_{(1-\beta,2s+s^*)}}\right)$.*

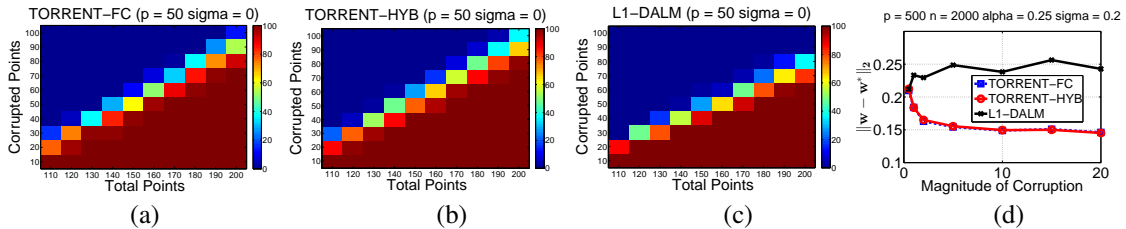

Figure 1: (a), (b) and (c) Phase-transition diagrams depicting the recovery properties of the TORRENT-FC, TORRENT-HYB and $L_1$ algorithms. The colors red and blue represent a high and low probability of success resp. A method is considered successful in an experiment if it recovers $\mathbf{w}^*$ upto a $10^{-4}$ relative error. Both variants of TORRENT can be seen to recover $\mathbf{w}^*$ in presence of larger number of corruptions than the $L_1$ solver. (d) Variation in recovery error with the magnitude of corruption. As the corruption is increased, TORRENT-FC and TORRENT-HYB show improved performance while the problem becomes more difficult for the $L_1$ solver.

*If $X$ also satisfies $\frac{4L_{(\beta,s+s^*)}}{\alpha_{(1-\beta,s+s^*)}} < 1$, then after $t = \mathcal{O}\left(\log\left(\frac{1}{\sqrt{n}}\frac{\|\mathbf{b}\|_2}{\epsilon}\right)\right)$ iterations, Algorithm 2 obtains an $\epsilon$-accurate solution $\mathbf{w}^t$ i.e. $\|\mathbf{w}^t - \mathbf{w}^*\|_2 \leq \epsilon$.*

*In particular, if $X$ is sampled from a Gaussian distribution $\mathcal{N}(\mathbf{0}, \Sigma)$ and $n \geq \Omega\left(s^* \cdot \frac{\lambda_{\max}(\Sigma)}{\lambda_{\min}(\Sigma)}\log p\right)$, then for all values of $\alpha \leq \beta < \frac{1}{65}$, we can guarantee $\|\mathbf{w}^t - \mathbf{w}^*\|_2 \leq \epsilon$ after $t = \mathcal{O}\left(\log\left(\frac{1}{\sqrt{n}}\frac{\|\mathbf{b}\|_2}{\epsilon}\right)\right)$ iterations of the algorithm (w.p. $\geq 1 - 1/n^{10}$).*

*Remark* 5. The sample complexity required by Theorem 9 is identical to the one required by analyses for high dimensional sparse recovery [12], save constants. Also note that TORRENT-HD can tolerate the same corruption index as TORRENT-FC.

## 6 Experiments

Several numerical simulations were carried out on linear regression problems in low-dimensional, as well as sparse high-dimensional settings. The experiments show that TORRENT not only offers statistically better recovery properties as compared to $L_1$-style approaches, but that it can be more than an order of magnitude faster as well.

**Data**: For the low dimensional setting, the regressor $\mathbf{w}^* \in \mathbb{R}^p$ was chosen to be a random unit norm vector. Data was sampled as $\mathbf{x_i} \sim \mathcal{N}(0, I_p)$ and response variables were generated as $y_i^* = \langle \mathbf{w}^*, \mathbf{x}_i \rangle$. The set of corrupted points $\overline{S}_*$ was selected as a uniformly random $(\alpha n)$-sized subset of $[n]$ and the corruptions were set to $b_i \sim U\left(-5\|\mathbf{y}^*\|_\infty, 5\|\mathbf{y}^*\|_\infty\right)$ for $i \in \overline{S}_*$. The corrupted responses were then generated as $y_i = y_i^* + b_i + \varepsilon_i$ where $\varepsilon_i \sim \mathcal{N}(0, \sigma^2)$. For the sparse high-dimensional setting, $\text{supp}(\mathbf{w}^*)$ was selected to be a random $s^*$-sized subset of $[p]$. Phase-transition diagrams (Figure 1) were generated by repeating each experiment 100 times. For all other plots, each experiment was run over 20 random instances of the data and the plots were drawn to depict the mean results.

**Algorithms**: We compared various variants of our algorithm TORRENT to the regularized $L_1$ algorithm for robust regression [4, 5]. Note that the $L_1$ problem can be written as $\min_{\mathbf{z}} \|\mathbf{z}\|_1$ s.t.$A\mathbf{z} = \mathbf{y}$, where $A = \left[X^\top \ \frac{1}{\lambda}I_{m \times m}\right]$ and $\mathbf{z}^* = \left[\mathbf{w}^{*\top} \ \lambda\mathbf{b}^\top\right]^\top$. We used the Dual Augmented Lagrange Multiplier (DALM) $L_1$ solver implemented by [15] to solve the $L_1$ problem. We ran a fine tuned grid search over the $\lambda$ parameter for the $L_1$ solver and quoted the best results obtained from the search. In the low-dimensional setting, we compared the recovery properties of TORRENT-FC (Algorithm 2) and TORRENT-HYB (Algorithm 4) with the DALM-$L_1$ solver, while for the high-dimensional case, we compared TORRENT-HD against the DALM-$L_1$ solver. Both the $L_1$ solver, as well as our methods, were implemented in Matlab and were run on a single core 2.4GHz machine with 8 GB RAM.

**Choice of $L_1$-solver**: An extensive comparative study of various $L_1$ minimization algorithms was performed by [15] who showed that the DALM and Homotopy solvers outperform other counterparts both in terms of recovery properties, and timings. We extended their study to our observation model and found the DALM solver to be significantly better than the other $L_1$ solvers; see Figure 3 in the appendix. We also observed, similar to [15], that the Approximate Message Passing (AMP) solver diverges on our problem as the input matrix to the $L_1$ solver is a non-Gaussian matrix $A = [X^T \frac{1}{\lambda}I]$.

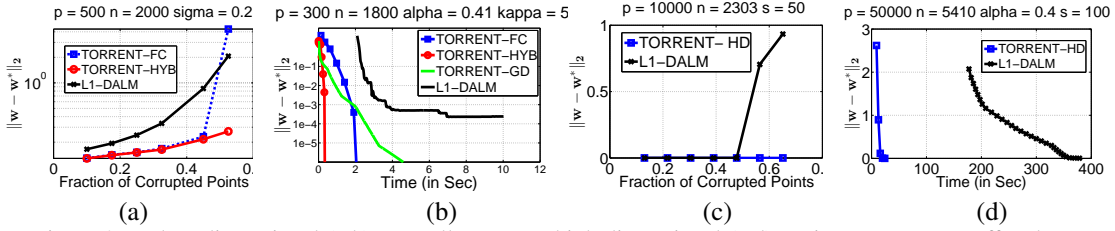

Figure 2: In low-dimensional (a,b), as well as sparse high dimensional (c,d) settings, TORRENT offers better recovery as the fraction of corrupted points $\alpha$ is varied. In terms of runtime, TORRENT is an order of magnitude faster than $L_1$ solvers in both settings. In the low-dim. setting, TORRENT-HYB is the fastest of all the variants.

**Evaluation Metric**: We measure the performance of various algorithms using the standard $L_2$ error: $r_{\widehat{\mathbf{w}}} = \|\widehat{\mathbf{w}} - \mathbf{w}^*\|_2$. For the phase-transition plots (Figure 1), we deemed an algorithm successful on an instance if it obtained a model $\widehat{\mathbf{w}}$ with error $r_{\widehat{\mathbf{w}}} < 10^{-4} \cdot \|\mathbf{w}^*\|_2$. We also measured the CPU time required by each of the methods, so as to compare their scalability.

## 6.1  Low Dimensional Results

**Recovery Property**: The phase-transition plots presented in Figure 1 represent our recovery experiments in graphical form. Both the fully-corrective and hybrid variants of TORRENT show better recovery properties than the $L_1$-minimization approach, indicated by the number of runs in which the algorithm was able to correctly recover $\mathbf{w}^*$ out of a 100 runs. Figure 2 shows the variation in recovery error as a function of $\alpha$ in the presence of white noise and exhibits the superiority of TORRENT-FC and TORRENT-HYB over $L_1$-DALM. Here again, TORRENT-FC and TORRENT-HYB achieve significantly lesser recovery error than $L_1$-DALM for all $\alpha <= 0.5$. Figure 3 in the appendix show that the variations of $\|\widehat{\mathbf{w}} - \mathbf{w}^*\|_2$ with varying $p, \sigma$ and $n$ follow a similar trend with TORRENT having significantly lower recovery error in comparison to the $L_1$ approach.

Figure 1(d) brings out an interesting trend in the recovery property of TORRENT. As we increase the magnitude of corruption from $U\left(-\|\mathbf{y}^*\|_\infty, \|\mathbf{y}^*\|_\infty\right)$ to $U\left(-20\|\mathbf{y}^*\|_\infty, 20\|\mathbf{y}^*\|_\infty\right)$, the recovery error for TORRENT-HYB and TORRENT-FC decreases as expected since it becomes easier to identify the grossly corrupted points. However the $L_1$-solver was unable to exploit this observation and in fact exhibited an increase in recovery error.

**Run Time**: In order to ascertain the recovery guarantees for TORRENT on ill-conditioned problems, we performed an experiment where data was sampled as $\mathbf{x}_i \sim \mathcal{N}(\mathbf{0}, \Sigma)$ where $\mathrm{diag}(\Sigma) \sim U(0, 5)$. Figure 2 plots the recovery error as a function of time. TORRENT-HYB was able to correctly recover $\mathbf{w}^*$ about $50\times$ faster than $L_1$-DALM which spent a considerable amount of time pre-processing the data matrix $X$. Even after allowing the $L_1$ algorithm to run for 500 iterations, it was unable to reach the desired residual error of $10^{-4}$. Figure 2 also shows that our TORRENT-HYB algorithm is able to converge to the optimal solution much faster than TORRENT-FC or TORRENT-GD. This is because TORRENT-FC solves a least square problem at each step and thus, even though it requires significantly fewer iterations to converge, each iteration in itself is very expensive. While each iteration of TORRENT-GD is cheap, it is still limited by the slow $\mathcal{O}\left((1 - \frac{1}{\kappa})^t\right)$ convergence rate of the gradient descent algorithm, where $\kappa$ is the condition number of the covariance matrix. TORRENT-HYB, on the other hand, is able to combine the strengths of both the methods to achieve faster convergence.

## 6.2  High Dimensional Results

**Recovery Property**: Figure 2 shows the variation in recovery error in the high-dimensional setting as the number of corrupted points was varied. For these experiments, $n$ was set to $5s^* \log(p)$ and the fraction of corrupted points $\alpha$ was varied from 0.1 to 0.7. While $L_1$-DALM fails to recover $\mathbf{w}^*$ for $\alpha > 0.5$, TORRENT-HD offers perfect recovery even for $\alpha$ values upto 0.7.

**Run Time**: Figure 2 shows the variation in recovery error as a function of run time in this setting. $L_1$-DALM was found to be an order of magnitude slower than TORRENT-HD, making it infeasible for sparse high-dimensional settings. One key reason for this is that the $L_1$-DALM solver is significantly slower in identifying the set of clean points. For instance, whereas TORRENT-HD was able to identify the clean set of points in only 5 iterations, it took $L_1$ around 250 iterations to do the same.

## Footnotes

*This work was done while P.K. was a postdoctoral researcher at Microsoft Research India.

[1]Note that for an adaptive adversary, as is the case in our work, recovery cannot be guaranteed for $\alpha \geq 1/2$ since the adversary can introduce corruptions as $\mathbf{b}_i = \mathbf{x}_i^\top (\widetilde{\mathbf{w}} - \mathbf{w}^*)$ for an adversarially chosen model $\widetilde{\mathbf{w}}$. This would make it impossible for any algorithm to distinguish between $\mathbf{w}^*$ and $\widetilde{\mathbf{w}}$ thus making recovery impossible.

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
