[Supplementary Material]

# A  Convergence Guarantees with Dense Noise and Sparse Corruptions

We will now present recovery guarantees for the TORRENT-FC algorithm when both, dense noise, as well as sparse adversarial corruptions are present. Extensions for TORRENT-GD and TORRENT-HYB will follow similarly.

**Theorem 10.** *Let $X = [\mathbf{x}_1, \dots, \mathbf{x}_n] \in \mathbb{R}^{p \times n}$ be the given data matrix and $\mathbf{y} = X^T \mathbf{w}^* + \mathbf{b} + \varepsilon$ be the corrupted output with sparse corruptions $\|\mathbf{b}\|_0 \leq \alpha \cdot n$ as well as dense bounded noise $\varepsilon$. Let Algorithm 2 be executed on this data with the thresholding parameter set to $\beta \geq \alpha$. Let $\Sigma_0$ be an invertible matrix such that $\widetilde{X} = \Sigma_0^{-1/2} X$ satisfies the SSC and SSS properties at level $\gamma$ with constants $\lambda_\gamma$ and $\Lambda_\gamma$ respectively (see Definition 1). If the data satisfies $\frac{4\sqrt{\Lambda_\beta}}{\sqrt{\lambda_{1-\beta}}} < 1$, then after $t = \mathcal{O}\left(\log\left(\frac{1}{\sqrt{n}}\frac{\|\mathbf{b}\|_2}{\epsilon}\right)\right)$ iterations, Algorithm 2 obtains an $\epsilon$-accurate solution $\mathbf{w}^t$ i.e. $\|\mathbf{w}^t - \mathbf{w}^*\|_2 \leq \epsilon + C\frac{\|\varepsilon\|_2}{\sqrt{n}}$ for some constant $C > 0$.*

*Proof.* We being by observing that the optimality of the model $\mathbf{w}^{t+1}$ on the active set $S_t$ ensures

$$\left\|\mathbf{y}_{S_t} - X_{S_t}^\top \mathbf{w}^{t+1}\right\|_2 = \left\|X_{S_t}^\top(\mathbf{w}^* - \mathbf{w}^{t+1}) + \varepsilon_{S_t} + \mathbf{b}_{S_t}\right\|_2 \leq \left\|\mathbf{y}_t - X_{S_t}^\top \mathbf{w}^*\right\|_2 = \left\|\varepsilon_{S_t} + \mathbf{b}_{S_t}\right\|_2,$$

which, upon the application of the triangle inequality, gives us

$$\left\|X_{S_t}^\top(\mathbf{w}^* - \mathbf{w}^{t+1})\right\|_2 \leq 2\left\|\varepsilon_{S_t} + \mathbf{b}_{S_t}\right\|_2.$$

Since $\left\|X_{S_t}^\top(\mathbf{w}^* - \mathbf{w}^{t+1})\right\|_2 \geq \sqrt{\lambda_{1-\beta}}\left\|\mathbf{w}^* - \mathbf{w}^{t+1}\right\|_2$, we get

$$\left\|\mathbf{w}^* - \mathbf{w}^{t+1}\right\|_2 \leq \frac{2}{\sqrt{\lambda_{1-\beta}}}\left\|\varepsilon_{S_t} + \mathbf{b}_{S_t}\right\|_2 \leq \frac{2}{\sqrt{\lambda_{1-\beta}}}\left(\|\varepsilon\|_2 + \|\mathbf{b}_{S_t}\|_2\right).$$

The hard thresholding step, on the other hand, guarantees that

$$\left\|X_{S_{t+1}}^\top(\mathbf{w}^* - \mathbf{w}^{t+1}) + \varepsilon_{S_{t+1}} + \mathbf{b}_{S_{t+1}}\right\|_2^2 = \left\|\mathbf{y}_{S_{t+1}} - X_{S_{t+1}}^\top \mathbf{w}^{t+1}\right\|_2^2$$
$$\leq \left\|\mathbf{y}_{S_*} - X_{S_*}^\top \mathbf{w}^{t+1}\right\|_2$$
$$= \left\|X_{S_*}^\top(\mathbf{w}^* - \mathbf{w}^{t+1}) + \varepsilon_{S_*}\right\|_2^2.$$

As before, let $\mathrm{CR}_{t+1} = S_{t+1}\backslash S_*$ and $\mathrm{MD}_{t+1} = S_*\backslash S_{t+1}$. Then we have

$$\left\|X_{\mathrm{CR}_{t+1}}^\top(\mathbf{w}^* - \mathbf{w}^{t+1}) + \varepsilon_{\mathrm{CR}_{t+1}} + \mathbf{b}_{\mathrm{CR}_{t+1}}\right\|_2 \leq \left\|X_{\mathrm{MD}_{t+1}}^\top(\mathbf{w}^* - \mathbf{w}^{t+1}) + \varepsilon_{\mathrm{MD}_{t+1}}\right\|_2.$$

An application of the triangle inequality and the fact that $\left\|\mathbf{b}_{\mathrm{CR}_{t+1}}\right\|_2 = \left\|\mathbf{b}_{S_{t+1}}\right\|$ gives us

$$\left\|\mathbf{b}_{S_{t+1}}\right\|_2 \leq \left\|X_{\mathrm{MD}_{t+1}}^\top(\mathbf{w}^* - \mathbf{w}^{t+1})\right\|_2 + \left\|X_{\mathrm{CR}_{t+1}}^\top(\mathbf{w}^* - \mathbf{w}^{t+1})\right\|_2 + \left\|\varepsilon_{\mathrm{CR}_{t+1}}\right\|_2 + \left\|\varepsilon_{\mathrm{MD}_{t+1}}\right\|_2$$
$$\leq 2\sqrt{\Lambda_\beta}\left\|\mathbf{w}^* - \mathbf{w}^{t+1}\right\|_2 + \sqrt{2}\left\|\varepsilon\right\|_2,$$
$$= \frac{4\sqrt{\Lambda_\beta}}{\sqrt{\lambda_{1-\beta}}}\left\|\mathbf{b}_{S_t}\right\|_2 + \left(\frac{4\sqrt{\Lambda_\beta}}{\sqrt{\lambda_{1-\beta}}} + \sqrt{2}\right)\left\|\varepsilon\right\|_2$$
$$\leq \eta \cdot \left\|\mathbf{b}_{S_t}\right\|_2 + (1 + \sqrt{2})\left\|\varepsilon\right\|_2,$$

where the second step uses the fact that $\max\left\{|\mathrm{CR}_{t+1}|, |\mathrm{MD}_{t+1}|\right\} \leq \beta \cdot n$ and the Cauchy-Schwartz inequality, and the last step uses the fact that for sufficiently small $\beta$, we have $\eta := \frac{4\sqrt{\Lambda_\beta}}{\sqrt{\lambda_{1-\beta}}}$. Using the inequality for $\left\|\mathbf{w}^{t+1} - \mathbf{w}^*\right\|_2$ again gives us

$$\left\|\mathbf{w}^* - \mathbf{w}^{t+1}\right\|_2 \leq \frac{2}{\sqrt{\lambda_{1-\beta}}}\left(\|\varepsilon\|_2 + \|\mathbf{b}_{S_t}\|_2\right)$$
$$\leq \frac{4 + 2\sqrt{2}}{\sqrt{\lambda_{1-\beta}}}\|\varepsilon\|_2 + \frac{2 \cdot \eta^t}{\sqrt{\lambda_{1-\beta}}}\|\mathbf{b}\|_2$$

For large enough $n$ we have $\sqrt{\lambda_{1-\beta}} \geq \mathcal{O}(\sqrt{n})$, which completes the proof. ☐

Notice that for random Gaussian noise, this result gives the following convergence guarantee.

**Corollary 11.** *Let the date be generated as before with random Gaussian dense noise i.e.* $\mathbf{y} = X^T\mathbf{w}^* + \mathbf{b} + \boldsymbol{\varepsilon}$ *with* $\|\mathbf{b}\|_0 \leq \alpha \cdot n$ *and* $\boldsymbol{\varepsilon} \sim \mathcal{N}(\mathbf{0}, \sigma^2 \cdot I)$. *Let Algorithm 2 be executed on this data with the thresholding parameter set to* $\beta \geq \alpha$. *Let* $\Sigma_0$ *be an invertible matrix such that* $\widetilde{X} = \Sigma_0^{-1/2}X$ *satisfies the SSC and SSS properties at level* $\gamma$ *with constants* $\lambda_\gamma$ *and* $\Lambda_\gamma$ *respectively (see Definition 1). If the data satisfies* $\frac{4\sqrt{\Lambda_\beta}}{\sqrt{\lambda_{1-\beta}}} < 1$, *then after* $t = \mathcal{O}\left(\log\left(\frac{1}{\sqrt{n}}\frac{\|\mathbf{b}\|_2}{\epsilon}\right)\right)$ *iterations, Algorithm 2 obtains an* $\epsilon$-*accurate solution* $\mathbf{w}^t$ *i.e.* $\|\mathbf{w}^t - \mathbf{w}^*\|_2 \leq \epsilon + 2\sigma C$, *where* $C > 0$ *is the constant in Theorem 10.*

*Proof.* Using tail bounds on Chi-squared distributions [16], we get, with probability at least $1 - \delta$,

$$\|\varepsilon\|_2^2 \leq \sigma^2\left(n + 2\sqrt{n\log\frac{1}{\delta}} + 2\log\frac{1}{\delta}\right).$$

Thus, for $n > 4\log\frac{1}{\delta}$, we have $\|\varepsilon\|_2^2 \leq 2\sigma n$ which proves the result. $\qquad\square$

*Remark* 6. We note that the design assumptions made by Theorem 10 (i..e $\frac{4\sqrt{\Lambda_\beta}}{\sqrt{\lambda_{1-\beta}}} < 1$) are similar to those made by Theorem 3 and would be satisfied with high probability by data sampled from sub-Gaussian distributions (see Appendix G for details).

*Remark* 7. We also note that Corollary 11 does not guarantee a consistent estimate of $\mathbf{w}^*$ whereas the least squares estimate is a consistent one in the non-corrupted regression setting. This is indeed a point of interest. However, we notice that existing works [6, 5] also are unable to tolerate a high level of adversarial corruption along with dense noise. Whereas [6] are only able to tolerate a vanishing $1/\sqrt{p}$ fraction of corruptions, [5] require the corruptions not be adaptive and be added independently of the data points and the white noise.

# B   Proof of Theorem 3

*Theorem 3.* Let $X = [\mathbf{x}_1, \ldots, \mathbf{x}_n] \in \mathbb{R}^{p\times n}$ be the given data matrix and $\mathbf{y} = X^T\mathbf{w}^* + \mathbf{b}$ be the corrupted output with $\|\mathbf{b}\|_0 \leq \alpha \cdot n$. Let Algorithm 2 be executed on this data with the thresholding parameter set to $\beta \geq \alpha$. Let $\Sigma_0$ be an invertible matrix such that $\widetilde{X} = \Sigma_0^{-1/2}X$ satisfies the SSC and SSS properties at level $\gamma$ with constants $\lambda_\gamma$ and $\Lambda_\gamma$ respectively (see Definition 1). If the data satisfies $\frac{(1+\sqrt{2})\Lambda_\beta}{\lambda_{1-\beta}} < 1$, then after $t = \mathcal{O}\left(\log\left(\frac{1}{\sqrt{n}}\frac{\|\mathbf{b}\|_2}{\epsilon}\right)\right)$ iterations, Algorithm 2 obtains an $\epsilon$-accurate solution $\mathbf{w}^t$ i.e. $\|\mathbf{w}^t - \mathbf{w}^*\|_2 \leq \epsilon$.

*Proof.* Let $\mathbf{r}^t = \mathbf{y} - X^\top\mathbf{w}^t$ be the vector of residuals at time $t$ and $C_t = X_{S_t}X_{S_t}^\top$. Since $\lambda_\alpha > 0$ (something which we shall establish later), we get

$$\mathbf{w}^{t+1} = C_t^{-1}X_{S_t}\mathbf{y}_{S_t} = C_t^{-1}X_{S_t}\left(X_{S_t}^\top\mathbf{w}^* + \mathbf{b}_{S_t}\right) = \mathbf{w}^* + C_t^{-1}X_{S_t}\mathbf{b}_{S_t}.$$

Thus, for any set $S \subset [n]$, we have

$$\mathbf{r}_S^{t+1} = \mathbf{y}_S - X_S^\top\mathbf{w}_{t+1} = \mathbf{b}_S - X_S^\top C_t^{-1}X_{S_t}\mathbf{b}_{S_t}$$

This, gives us

$$\left\|\mathbf{b}_{S_{t+1}}\right\|_2^2 = \left\|\mathbf{b}_{S_{t+1}} - X_{S_{t+1}}^\top C_t^{-1}X_{S_t}\mathbf{b}_{S_t}\right\|_2^2 - \left\|X_{S_{t+1}}^\top C_t^{-1}X_{S_t}\mathbf{b}_{S_t}\right\|_2^2 + 2\cdot\mathbf{b}_{S_{t+1}}^\top X_{S_{t+1}}^\top C_t^{-1}X_{S_t}\mathbf{b}_{S_t}$$

$$\overset{\zeta_1}{\leq} \left\|\mathbf{b}_{S_*} - X_{S_*}^\top C_t^{-1}X_{S_t}\mathbf{b}_{S_t}\right\|_2^2 - \left\|X_{S_{t+1}}^\top C_t^{-1}X_{S_t}\mathbf{b}_{S_t}\right\|_2^2 + 2\cdot\mathbf{b}_{S_{t+1}}^\top X_{S_{t+1}}^\top C_t^{-1}X_{S_t}\mathbf{b}_{S_t}$$

$$\overset{\zeta_2}{=} \left\|X_{S_*}^\top C_t^{-1}X_{S_t}\mathbf{b}_{S_t}\right\|_2^2 - \left\|X_{S_{t+1}}^\top C_t^{-1}X_{S_t}\mathbf{b}_{S_t}\right\|_2^2 + 2\cdot\mathbf{b}_{S_{t+1}}^\top X_{S_{t+1}}^\top C_t^{-1}X_{S_t}\mathbf{b}_{S_t}$$

$$\leq \left\|X_{S_*\backslash S_{t+1}}^\top C_t^{-1}X_{S_t}\mathbf{b}_{S_t}\right\|_2^2 + 2\cdot\mathbf{b}_{S_{t+1}}^\top X_{S_{t+1}}^\top C_t^{-1}X_{S_t}\mathbf{b}_{S_t}$$

$$\stackrel{\zeta_3}{\equiv} \left\| \widetilde{X}_{S_* \setminus S_{t+1}}^\top \left( \widetilde{X}_{S_t} \widetilde{X}_{S_t}^T \right)^{-1} \widetilde{X}_{S_t} \mathbf{b}_{S_t} \right\|_2^2 + 2 \cdot \mathbf{b}_{S_{t+1}}^\top \widetilde{X}_{S_{t+1}}^\top \left( \widetilde{X}_{S_t} \widetilde{X}_{S_t}^T \right)^{-1} \widetilde{X}_{S_t} \mathbf{b}_{S_t}$$

$$\stackrel{\zeta_4}{\leq} \frac{\Lambda_\beta^2}{\lambda_{1-\beta}^2} \cdot \|\mathbf{b}_{S_t}\|_2^2 + 2 \cdot \frac{\Lambda_\beta}{\lambda_{1-\beta}} \cdot \|\mathbf{b}_{S_t}\|_2 \|\mathbf{b}_{S_{t+1}}\|_2 \,,$$

where $\zeta_1$ follows since the hard thresholding step ensures $\left\| \mathbf{r}_{S_{t+1}}^{t+1} \right\|_2^2 \leq \left\| \mathbf{r}_{S_*}^{t+1} \right\|_2^2$ (see Claim 19 and use the fact that $\beta \geq \alpha$), $\zeta_2$ notices the fact that $\mathbf{b}_{S_*} = \mathbf{0}$. $\zeta_3$ follows from setting $\widetilde{X} = \Sigma_0^{-1/2} X$ and $X_S^\top C_t^{-1} X_{S'} = \widetilde{X}_S^\top (\widetilde{X}_{S_t} \widetilde{X}_{S_t}^\top)^{-1} \widetilde{X}_{S'}$. $\zeta_4$ follows from the definition of SSC and SSS properties, $\|\mathbf{b}_{S_t}\|_0 \leq \|\mathbf{b}\|_0 \leq \beta \cdot n$ and $|S_* \setminus S_{t+1}| \leq \beta \cdot n$. Solving the quadratic equation gives us

$$\left\| \mathbf{b}_{S_{t+1}} \right\|_2 \leq (1 + \sqrt{2}) \cdot \frac{\Lambda_\beta}{\lambda_{1-\beta}} \cdot \|\mathbf{b}_{S_t}\|_2 \,. \tag{4}$$

Let $\eta := \frac{(1+\sqrt{2})\Lambda_\beta}{\lambda_{1-\beta}}$ denote the convergence rate in (4). We shall show below that for a large family of random designs, we have $\eta < 1$ if $n \geq \Omega\left(p + \log \frac{1}{\delta}\right)$. We now recall from our earlier discussion that $\mathbf{w}^{t+1} = \mathbf{w}^* + C_t^{-1} X_{S_t} \mathbf{b}_{S_t}$ which gives us

$$\left\| \mathbf{w}^{t+1} - \mathbf{w}^* \right\|_2 = \left\| C_t^{-1} X_{S_t} \mathbf{b}_{S_t} \right\|_2 \leq \frac{\sqrt{\Lambda_\beta}}{\lambda_{1-\beta}} \cdot \|\mathbf{b}_{S_t}\|_2 \leq \eta^t \cdot \frac{\sqrt{\Lambda_\beta}}{\lambda_{1-\beta}} \|\mathbf{b}\|_2 \leq \epsilon,$$

for $t \geq \log_{\frac{1}{\eta}}\left( \frac{\sqrt{\Lambda_\beta}}{\lambda_{1-\beta}} \cdot \frac{\|\mathbf{b}\|_2}{\epsilon} \right)$. Noting that $\frac{\sqrt{\Lambda_\beta}}{\lambda_{1-\beta}} \leq \mathcal{O}\left( \frac{1}{\sqrt{n}} \right)$ establishes the convergence result. $\quad\square$

## C Proof of Theorem 4

*Theorem 4.* Let $X = [\mathbf{x}_1, \ldots, \mathbf{x}_n] \in \mathbb{R}^{p \times n}$ be the given data matrix with each $\mathbf{x}_i \sim \mathcal{N}(\mathbf{0}, \Sigma)$. Let $\mathbf{y} = X^\top \mathbf{w}^* + \mathbf{b}$ and $\|\mathbf{b}\|_0 \leq \alpha \cdot n$. Also, let $\alpha \leq \beta < \frac{1}{65}$ and $n \geq \Omega\left(p + \log \frac{1}{\delta}\right)$. Then, with probability at least $1 - \delta$, the data satisfies $\frac{(1+\sqrt{2})\Lambda_\beta}{\lambda_{1-\beta}} < \frac{9}{10}$. More specifically, after $T \geq 10 \log\left( \frac{1}{\sqrt{n}} \frac{\|\mathbf{b}\|_2}{\epsilon} \right)$ iterations of Algorithm 1 with the thresholding parameter set to $\beta$, we have $\left\| \mathbf{w}^T - \mathbf{w}^* \right\| \leq \epsilon$.

*Proof.* We note that whenever $\mathbf{x} \sim \mathcal{N}(\mathbf{0}, \Sigma)$ then $\Sigma^{-1/2} \mathbf{x} \sim \mathcal{N}(\mathbf{0}, I)$. Thus, Theorem 15 assures us that with probability at least $1 - \delta$, the data matrix $\widetilde{X} = \Sigma^{-1/2} X$ satisfies the SSC and SSS properties with the following constants

$$\Lambda_\beta \leq \beta n \left( 1 + 3e\sqrt{6 \log \frac{e}{\beta}} \right) + \mathcal{O}\left( \sqrt{np + n \log \frac{1}{\delta}} \right)$$

$$\lambda_{1-\beta} \geq n - \beta n \left( 1 + 3e\sqrt{6 \log \frac{e}{\beta}} \right) - \Omega\left( \sqrt{np + n \log \frac{1}{\delta}} \right)$$

Thus, the convergence given be Algorithm 1, when invoked with $\Sigma_0 = \Sigma$, relies on the quantity $\eta = \frac{(1+\sqrt{2})\Lambda_\beta}{\lambda_{1-\beta}}$ being less than unity. This translates to the requirement $(1 + \sqrt{2})\Lambda_\beta \leq \lambda_{1-\beta}$. Using the above bounds translates that requirement to

$$\underbrace{(2 + \sqrt{2})\beta \left( 1 + 3e\sqrt{6 \log \frac{e}{\beta}} \right)}_{(A)} + \underbrace{\mathcal{O}\left( \sqrt{\frac{p}{n} + \frac{1}{n} \log \frac{1}{\delta}} \right)}_{(B)} < 1.$$

For $n = \Omega\left(p + \log \frac{1}{\delta}\right)$, the second quantity $(B)$ can be made as small a constant as necessary. Tackling the first quantity $(A)$ turns out to be more challenging. However, we can show that for all $\beta < \frac{1}{190}$, we get $\eta = \frac{(1+\sqrt{2})\Lambda_\beta}{\lambda_{1-\beta}} < \frac{9}{10}$ which establishes the claimed result. Thus, Algorithm 1 can tolerate a corruption index of upto $\alpha \leq \frac{1}{190}$. However, we note that using a more finely tuned setting of the constant $\epsilon$ in the proof of Theorem 15 and a more careful proof using tight tail inequalities for chi-squared distributions [16], we can achieve a better corruption level tolerance of $\alpha < \frac{1}{65}$. The constants in the expression for $n$ can be optimized as well. The current bound can be shown to hold for $n \geq 270\left(p + \log \frac{1}{\delta}\right)$. $\quad\square$

# D Proof of Theorem 5

*Theorem 5.* Let $X = [\mathbf{x}_1, \ldots, \mathbf{x}_n] \in \mathbb{R}^{p \times n}$ be the given data matrix and $\mathbf{y} = X^T \mathbf{w}^* + \mathbf{b}$ be the corrupted output with $\|\mathbf{b}\|_0 \leq \alpha \cdot n$. Let $X$ satisfy the SSC and SSS properties at level $\gamma$ with constants $\lambda_\gamma$ and $\Lambda_\gamma$ respectively (see Definition 1). Let Algorithm 1 be executed on this data with the GD update (Algorithm 3) with the thresholding parameter set to $\beta \geq \alpha$ and the step length set to $\eta = \frac{1}{\Lambda_{1-\beta}}$. If the data satisfies $\max\left\{\eta\sqrt{\Lambda_\beta}, 1 - \eta\lambda_{1-\beta}\right\} \leq \frac{1}{4}$, then after $t = \mathcal{O}\left(\log\left(\frac{\|b\|_2}{\sqrt{n}}\frac{1}{\epsilon}\right)\right)$ iterations, Algorithm 1 obtains an $\epsilon$-accurate solution $\mathbf{w}^t$ i.e. $\|\mathbf{w}^t - \mathbf{w}^*\|_2 \leq \epsilon$.

*Proof.* Let $\mathbf{r}^t = \mathbf{y} - X^\top \mathbf{w}^t$ be the vector of residuals at time $t$ and $C_t = X_{S_t} X_{S_t}^\top$. We have

$$\mathbf{w}^{t+1} = \mathbf{w}^t + \eta \cdot X_{S_t} \mathbf{r}_{S_t}^t = \mathbf{w}^t + \eta \cdot X_{S_t}(\mathbf{y}_{S_t} - X_{S_t}^\top \mathbf{w}^t)$$

The thresholding step ensures that $\left\|\mathbf{r}_{S_{t+1}}^{t+1}\right\|_2^2 \leq \left\|\mathbf{r}_{S_*}^{t+1}\right\|_2^2$ (see Claim 19 and use $\beta \geq \alpha$) which implies

$$\left\|\mathbf{r}_{\text{CR}_{t+1}}^{t+1}\right\|_2^2 \leq \left\|\mathbf{r}_{\text{MD}_{t+1}}^{t+1}\right\|_2^2,$$

where $\text{CR}_{t+1} = S_{t+1} \setminus S_*$ are the *corrupted recoveries* and $\text{MD}_{t+1} = S_* \setminus S_{t+1}$ are the clean points *missed* out from *detection*. Note that $|\text{CR}_{t+1}| \leq \alpha \cdot n$ and $|\text{MD}_{t+1}| \leq \beta \cdot n$. Since $\mathbf{b}_{S_*} = \mathbf{0}$ and $\text{MD}_{t+1} \subseteq S_*$, we get

$$\left\|\mathbf{b}_{\text{CR}_{t+1}} + X_{\text{CR}_{t+1}}^\top(\mathbf{w}^* - \mathbf{w}^{t+1})\right\|_2 \leq \left\|X_{\text{MD}_{t+1}}^\top(\mathbf{w}^* - \mathbf{w}^{t+1})\right\|_2$$

Using the SSS conditions and the fact that $\|\mathbf{b}_{S_{t+1}}\|_2 = \|\mathbf{b}_{S_{t+1} \setminus S_*}\|_2$ gives us

$$\|\mathbf{b}_{S_{t+1}}\|_2 = \|\mathbf{b}_{\text{CR}_{t+1}}\|_2 \leq (\sqrt{\Lambda_\alpha} + \sqrt{\Lambda_\beta})\|\mathbf{w}^* - \mathbf{w}^{t+1}\|_2 \leq 2\sqrt{\Lambda_\beta}\|\mathbf{w}^* - \mathbf{w}^{t+1}\|_2$$

Now, using the expression for $\mathbf{w}^{t+1}$ gives us

$$\|\mathbf{w}^* - \mathbf{w}^{t+1}\|_2 \leq \|(I - \eta C_t)(\mathbf{w}^* - \mathbf{w}^t)\|_2 + \eta \|X_{S_t}\mathbf{b}_{S_t}\|_2$$

We will bound the two terms on the right hand separately. We can bound the second term easily as

$$\eta \|X_{S_t}\mathbf{b}_{S_t}\|_2 \leq \eta\sqrt{\Lambda_\alpha}\|\mathbf{b}_{S_t}\|_2 \leq \eta\sqrt{\Lambda_\beta}\|\mathbf{b}_{S_t}\|_2,$$

since $\|\mathbf{b}_{S_t}\|_0 \leq \alpha \cdot n$. For the first term we observe that for $\eta \leq \frac{1}{\Lambda_{1-\beta}}$, we have

$$\|I - \eta C_t\|_2 = \sup_{\mathbf{v} \in S^{p-1}} |1 - \eta \cdot \mathbf{v}^\top C_t \mathbf{v}| = \sup_{\mathbf{v} \in S^{p-1}} \left\{1 - \eta \cdot \mathbf{v}^\top C_t \mathbf{v}\right\} \leq 1 - \eta\lambda_{1-\beta},$$

which we can use to bound

$$\|\mathbf{w}^* - \mathbf{w}^{t+1}\|_2 \leq (1 - \eta\lambda_{1-\beta})\|\mathbf{w}^* - \mathbf{w}^t\|_2 + \eta\sqrt{\Lambda_\beta}\|\mathbf{b}_{S_t}\|_2$$

This gives us, for $\eta = \frac{1}{\Lambda_{1-\beta}}$,

$$\|\mathbf{b}_{S_{t+1}}\|_2 \leq 2\sqrt{\Lambda_\beta}\|\mathbf{w}^* - \mathbf{w}^{t+1}\|_2 \leq 2\underbrace{\left(1 - \frac{\lambda_{1-\beta}}{\Lambda_{1-\beta}}\right)}_{(P)}\sqrt{\Lambda_\beta}\|\mathbf{w}^* - \mathbf{w}^t\|_2 + 2\underbrace{\frac{\Lambda_\beta}{\Lambda_{1-\beta}}}_{(Q)}\|\mathbf{b}_{S_t}\|_2.$$

For Gaussian designs and small enough $\beta$, we can show $(Q) \leq \frac{1}{4}$ as we did in Theorem 4. To bound $(P)$, we use the lower bound on $\lambda_{1-\beta}$ given by Theorem 15 and use the following tighter upper bound for $\Lambda_{1-\beta}$:

$$\Lambda_{1-\beta} \leq \left((1 - \beta) + 3e\sqrt{6\beta(1-\beta)\log\frac{e}{\beta}}\right)n + \mathcal{O}\left(\sqrt{np + n\log\frac{1}{\delta}}\right)$$

The above bound is obtained similarly to the one in Theorem 15 but uses the identity $\binom{n}{k} = \binom{n}{n-k} \leq \left(\frac{en}{n-k}\right)^{n-k}$ for values of $k \geq n/2$ instead. For small enough $\beta$ and $n = \Omega\left(\kappa^2(\Sigma)(p + \log\frac{1}{\delta})\right)$, we can then show $(P) \leq \frac{1}{4}$ as well. Let $\Psi_t := \sqrt{n}\|\mathbf{w}^* - \mathbf{w}^t\|_2 + \|b_{S_t}\|$. Using elementary manipulations and the fact that $\sqrt{\Lambda_\beta} \geq \Omega(\sqrt{n})$, we can then show that

$$\Psi_{t+1} \leq 3/4 \cdot \Psi_t.$$

Thus, in $t = \mathcal{O}\left(\log\left(\left(\|\mathbf{w}^*\|_2 + \frac{\|b\|_2}{\sqrt{n}}\right)\frac{1}{\epsilon}\right)\right)$ iterations of the algorithm, we arrive at an $\epsilon$-optimal solution i.e. $\|\mathbf{w}^* - \mathbf{w}^t\|_2 \leq \epsilon$. A similar argument holds true for sub-Gaussian designs as well. $\quad\square$

# E  Proof of Theorem 6

*Theorem 6.* Suppose Algorithm 4 is executed on data that allows Algorithms 2 and 3 a convergence rate of $\eta_{\text{FC}}$ and $\eta_{\text{GD}}$ respectively. Suppose we have $2 \cdot \eta_{\text{FC}} \cdot \eta_{\text{GD}} < 1$. Then for *any* interleavings of the FC and GD steps that the policy may enforce, after $t = \mathcal{O}\left(\log\left(\frac{1}{\sqrt{n}}\frac{\|\mathbf{b}\|_2}{\epsilon}\right)\right)$ iterations, Algorithm 4 ensures an $\epsilon$-optimal solution i.e. $\|\mathbf{w}^t - \mathbf{w}^*\| \leq \epsilon$.

*Proof.* Our proof shall essentially show that the FC and GD steps do not undo the progress made by the other if executed in succession and if $2 \cdot \eta_{\text{FC}} \cdot \eta_{\text{GD}} < 1$, actually ensure non-trivial progress. Let

$$\Psi_t^{\text{FC}} = \|\mathbf{b}_{S_t}\|_2$$
$$\Psi_t^{\text{GD}} = \sqrt{n}\left\|\mathbf{w}^t - \mathbf{w}^*\right\| + \|\mathbf{b}_{S_t}\|_2$$

denote the potential functions used in the analyses of the FC and GD algorithms before. Then we will show below that if the FC and GD algorithms are executed in steps $t$ and $t+1$ then we have

$$\Psi_{t+2}^{\text{FC}} \leq 2 \cdot \eta_{\text{FC}} \cdot \eta_{\text{GD}} \cdot \Psi_t^{\text{FC}}$$

Alternatively, if the GD and FC algorithms are executed in steps $t$ and $t+1$ respectively, then

$$\Psi_{t+2}^{\text{GD}} \leq 2 \cdot \eta_{\text{FC}} \cdot \eta_{\text{GD}} \cdot \Psi_t^{\text{GD}}$$

Thus, if algorithm executes the FC step at the time step $t$, then it would at least ensure $\Psi_t^{\text{FC}} \leq (2 \cdot \eta_{\text{FC}} \cdot \eta_{\text{GD}})^{t/2} \cdot \Psi_0^{\text{FC}}$ (similarly if the last step is a GD step). Since both the FC and GD algorithms ensure $\|\mathbf{w}^t - \mathbf{w}^*\|_2 \leq \epsilon$ for $t \geq \mathcal{O}\left(\log\left(\frac{1}{\sqrt{n}}\frac{\|b\|_2}{\epsilon}\right)\right)$, the claim would follow.

We now prove the two claimed results regarding the two types of interleaving below

1. FC $\longrightarrow$ GD
   The FC step guarantees $\left\|\mathbf{b}_{S_{t+1}}\right\|_2 \leq \eta_{\text{FC}} \cdot \|\mathbf{b}_{S_t}\|$ as well as $\left\|\mathbf{w}^{t+1} - \mathbf{w}^*\right\|_2 \leq \eta_{\text{FC}} \cdot \frac{\|\mathbf{b}_{S_t}\|}{\sqrt{n}}$, whereas the GD step guarantees $\Psi_{t+2}^{\text{GD}} \leq \eta_{\text{GD}} \cdot \Psi_{t+1}^{\text{GD}}$. Together these guarantee

$$\sqrt{n}\left\|\mathbf{w}^{t+2} - \mathbf{w}^*\right\|_2 + \left\|\mathbf{b}_{S_{t+2}}\right\|_2 \leq \eta_{\text{GD}} \cdot \sqrt{n}\left\|\mathbf{w}^{t+1} - \mathbf{w}^*\right\|_2 + \left\|\mathbf{b}_{S_{t+1}}\right\|_2$$
$$\leq 2 \cdot \eta_{\text{FC}} \cdot \eta_{\text{GD}} \cdot \|\mathbf{b}_{S_t}\|_2$$

   Since $\sqrt{n}\left\|\mathbf{w}^{t+2} - \mathbf{w}^*\right\|_2 \geq 0$, this yields the result.

2. GD $\longrightarrow$ FC
   The GD step guarantees $\Psi_{t+1}^{\text{GD}} \leq \eta_{\text{GD}} \cdot \Psi_t^{\text{GD}}$ whereas the FC step guarantees $\left\|\mathbf{b}_{S_{t+2}}\right\|_2 \leq \eta_{\text{FC}} \cdot \left\|\mathbf{b}_{S_{t+1}}\right\|$ as well as $\left\|\mathbf{w}^{t+2} - \mathbf{w}^*\right\|_2 \leq \eta_{\text{FC}} \cdot \frac{\|\mathbf{b}_{S_{t+1}}\|}{\sqrt{n}}$. Together these guarantee

$$\sqrt{n}\left\|\mathbf{w}^{t+2} - \mathbf{w}^*\right\|_2 + \left\|\mathbf{b}_{S_{t+2}}\right\|_2 \leq 2\eta_{\text{FC}}\left\|\mathbf{b}_{S_{t+1}}\right\|_2$$
$$\leq 2 \cdot \eta_{\text{FC}} \cdot \eta_{\text{GD}} \cdot \Psi_t^{\text{GD}},$$

   where the second step follows from the GD step guarantee since $\sqrt{n}\left\|\mathbf{w}^{t+1} - \mathbf{w}^*\right\|_2 \geq 0$.

This finishes the proof. $\qquad\square$

# F  Proof of Theorem 9

*Theorem 9.* Let $X = [\mathbf{x}_1, \ldots, \mathbf{x}_n] \in \mathbb{R}^{p \times n}$ be the given data matrix and $\mathbf{y} = X^T\mathbf{w}^* + \mathbf{b}$ be the corrupted output with $\|\mathbf{w}^*\|_0 \leq s^*$ and $\|\mathbf{b}\|_0 \leq \alpha \cdot n$. Let Algorithm 2 be executed on this data with the IHT update from [12] and thresholding parameter set to $\beta \geq \alpha$. Let $\Sigma_0$ be an invertible matrix such that $\Sigma_0^{-1/2}X$ satisfies the SRSC and SRSS properties at level $(\gamma, 2s+s^*)$ with constants $\alpha_{(\gamma,2s+s^*)}$ and $L_{(\gamma,2s+s^*)}$ respectively (see Definition 8) for $s \geq 32\left(\frac{L_{(\gamma,2s+s^*)}}{\alpha_{(\gamma,2s+s^*)}}\right)$ with $\gamma = 1 - \beta$. If $X$ also satisfies $\frac{4L_{(\beta,s+s^*)}}{\alpha_{(1-\beta,s+s^*)}} < 1$, then after $t = \mathcal{O}\left(\log\left(\frac{1}{\sqrt{n}}\frac{\|\mathbf{b}\|_2}{\epsilon}\right)\right)$ iterations, Algorithm 2 obtains

an $\epsilon$-accurate solution $\mathbf{w}^t$ i.e. $\left\| \mathbf{w}^t - \mathbf{w}^* \right\|_2 \leq \epsilon$. In particular, if $X$ is sampled from a Gaussian distribution $\mathcal{N}(\mathbf{0}, \Sigma)$ and $n \geq \Omega\left( (2s + s^*) \log p + \log \frac{1}{\delta} \right)$, then for all values of $\alpha \leq \beta < \frac{1}{65}$, we can guarantee recovery as $\left\| \mathbf{w}^t - \mathbf{w}^* \right\|_2 \leq \epsilon$.

*Proof.* We first begin with the guarantee provided by existing sparse recovery techniques. The results of [12], for example, indicate that if the input to the algorithm indeed satisfies the RSC and RSS properties at the level $(1 - \beta, 2s + s^*)$ with constants $\alpha_{2s+s^*}$ and $L_{2s+s^*}$ for $s \geq 32\left( \frac{L_{2s+s^*}}{\alpha_{2s+s^*}} \right)$, then in time $\tau = \mathcal{O}\left( \frac{L_{2s+s^*}}{\alpha_{2s+s^*}} \cdot \log\left( \frac{\|b\|_2}{\rho} \right) \right)$, the IHT algorithm [12, Algorithm 1] outputs an updated model $\mathbf{w}^{t+1}$ that satisfies $\left\| \mathbf{w}^{t+1} \right\|_0 \leq s$, as well as

$$\left\| \mathbf{y}_{S_t} - X_{S_t}^\top \mathbf{w}^{t+1} \right\|_2^2 \leq \left\| \mathbf{y}_{S_t} - X_{S_t}^\top \mathbf{w}^* \right\|_2^2 + \rho.$$

We will set $\rho$ later. Since the SRSC and SRSS properties ensure the above and $\mathbf{y} = X^\top \mathbf{w}^* + \mathbf{b}$, this gives us

$$\left\| X_{S_t}^\top (\mathbf{w}^{t+1} - \mathbf{w}^*) \right\|_2^2 \leq 2(\mathbf{w}^{t+1} - \mathbf{w}^*)^\top X_{S_t}^\top \mathbf{b}_{S_t} + \rho = 2(\mathbf{w}^{t+1} - \mathbf{w}^*)^\top X_{S_t \cap \bar{S}_*}^\top \mathbf{b}_{S_t \cap \bar{S}_*} + \rho,$$

since $\mathbf{b}_S = \mathbf{0}$ for any set $S \cap \bar{S}_* = \phi$. We now analyze the two sides separately below using the SRSC and SRSS properties below. For any $S \subset [n]$, denote $\tilde{X}_S := \Sigma_0^{-1/2} X$.

$$\left\| X_{S_t}^\top (\mathbf{w}^{t+1} - \mathbf{w}^*) \right\|_2^2 = \left\| \tilde{X}_{S_t}^\top \Sigma_0^{1/2} (\mathbf{w}^{t+1} - \mathbf{w}^*) \right\|_2^2 \geq \alpha_{(1-\beta, s+s^*)} \left\| \Sigma_0^{1/2} (\mathbf{w}^{t+1} - \mathbf{w}^*) \right\|_2^2$$

$$\left\| X_{S_t \cap \bar{S}_*} (\mathbf{w}^{t+1} - \mathbf{w}^*) \right\| = \left\| \tilde{X}_{S_t \cap \bar{S}_*} \Sigma_0^{1/2} (\mathbf{w}^{t+1} - \mathbf{w}^*) \right\| \leq \sqrt{L_{(\beta, s+s^*)}} \left\| \Sigma_0^{1/2} (\mathbf{w}^{t+1} - \mathbf{w}^*) \right\|_2.$$

Now, if $\left\| \mathbf{w}^{t+1} - \mathbf{w}^* \right\|_2 \geq \epsilon$, then $\left\| \Sigma_0^{1/2} (\mathbf{w}^{t+1} - \mathbf{w}^*) \right\|_2 \geq \sqrt{\lambda_{\min}(\Sigma_0)} \cdot \epsilon$. This give us

$$\left\| \Sigma_0^{1/2} (\mathbf{w}^{t+1} - \mathbf{w}^*) \right\|_2 \leq \frac{2\sqrt{L_{(\beta, s+s^*)}}}{\alpha_{(1-\beta, s+s^*)}} \left\| \mathbf{b}_{S_t \cap \bar{S}_*} \right\|_2 + \frac{\rho}{\alpha_{(1-\beta, s+s^*)}}$$

$$= \frac{2\sqrt{L_{(\beta, s+s^*)}}}{\alpha_{(1-\beta, s+s^*)}} \left\| \mathbf{b}_{S_t} \right\|_2 + \frac{\rho}{\epsilon \cdot \sqrt{\lambda_{\min}(\Sigma_0)} \cdot \alpha_{(1-\beta, s+s^*)}}.$$

We note that although we declared the SRSC and SRSS properties for the action of matrices on sparse vectors (such as $\mathbf{w}^* - \mathbf{w}^{t+1}$), we instead applied them above to the action of matrices on sparse vectors transformed by $\Sigma_0^{1/2}$ ($\Sigma_0^{1/2}(\mathbf{w}^* - \mathbf{w}^{t+1})$). Since $\Sigma_0^{1/2}\mathbf{v}$ need not be sparse even if $\mathbf{v}$ is sparse, this appears to pose a problem. However, all we need to resolve this is to notice that the proof technique of Theorem 18 which would be used to establish the SRSC and SRSS properties, holds in general for not just the action of a matrix on the set of sparse vectors, but on vectors in the union of any fixed set of low dimensional subspaces.

More specifically, we can modify the RSC and RSS properties (and by extension, the SRSC and SRSS properties), to requiring that the matrix $X$ act as an approximate isometry on the following set of vectors $S_{(s, \Sigma_0)}^{p-1} := \left\{ \mathbf{v} : \mathbf{v} = \Sigma_0^{-1/2}\mathbf{v}' \text{ for some } \mathbf{v}' \in S_s^{p-1} \right\}$. We refer the reader to the work of [17] which describes this technique in great detail. Proceeding with the proof, the assurance of the thresholding step, as used in the proof of Theorem 5, along with a straightforward application of the (modified) SRSS property gives us

$$\left\| \mathbf{b}_{S_{t+1}} \right\|_2 \leq \left\| X_{\mathrm{CR}_{t+1}}^\top (\mathbf{w}^{t+1} - \mathbf{w}^*) \right\|_2 + \left\| X_{\mathrm{MD}_{t+1}}^\top (\mathbf{w}^{t+1} - \mathbf{w}^*) \right\|_2$$

$$= \left\| \tilde{X}_{\mathrm{CR}_{t+1}}^\top \Sigma_0^{1/2} (\mathbf{w}^{t+1} - \mathbf{w}^*) \right\|_2 + \left\| \tilde{X}_{\mathrm{MD}_{t+1}}^\top \Sigma_0^{1/2} (\mathbf{w}^{t+1} - \mathbf{w}^*) \right\|_2$$

$$\leq 2\sqrt{L_{(\beta, s+s^*)}} \left\| \Sigma_0^{1/2} (\mathbf{w}^{t+1} - \mathbf{w}^*) \right\|_2$$

$$\leq \frac{4 L_{(\beta, s+s^*)}}{\alpha_{(1-\beta, s+s^*)}} \left\| \mathbf{b}_{S_t} \right\|_2 + \frac{2\rho \sqrt{L_{(\beta, s+s^*)}}}{\epsilon \cdot \sqrt{\lambda_{\min}(\Sigma_0)} \cdot \alpha_{(1-\beta, s+s^*)}}$$

Thus, whenever $\left\|\mathbf{w}^{t+1} - \mathbf{w}^*\right\|_2 > \epsilon$, in successive steps, $\left\|\mathbf{b}_{S_t}\right\|_2$ undergoes a linear decrease. Denoting $\eta := \frac{4L_{(\beta,s+s^*)}}{\alpha_{(1-\beta,s+s^*)}}$, we get

$$\left\|\mathbf{b}_{S_{t+1}}\right\|_2 \leq \eta^t \cdot \|\mathbf{b}\|_2 + \left(\frac{1-\eta^t}{1-\eta}\right) \frac{2\rho\sqrt{L_{(\beta,s+s^*)}}}{\epsilon \cdot \sqrt{\lambda_{\min}(\Sigma_0)} \cdot \alpha_{(1-\beta,s+s^*)}}$$

and using $\left\|\Sigma_0^{1/2}(\mathbf{w}^t - \mathbf{w}^*)\right\|_2 \geq \sqrt{\lambda_{\min}(\Sigma_0)} \|\mathbf{w}^t - \mathbf{w}^*\|_2$ gives us

$$\left\|\mathbf{w}^{t+1} - \mathbf{w}^*\right\|_2 \leq \frac{2\sqrt{L_{(\beta,s+s^*)}}}{\sqrt{\lambda_{\min}(\Sigma_0)} \cdot \alpha_{(1-\beta,s+s^*)}} \left\|\mathbf{b}_{S_{t+1}}\right\|_2 + \frac{\rho}{\lambda_{\min}(\Sigma_0) \cdot \alpha_{(1-\beta,s+s^*)}}$$

$$\leq \eta^t \frac{2\sqrt{L_{(\beta,s+s^*)}}}{\sqrt{\lambda_{\min}(\Sigma_0)} \cdot \alpha_{(1-\beta,s+s^*)}} \|\mathbf{b}\|_2 + \frac{36\rho}{\epsilon \cdot \lambda_{\min}(\Sigma_0) \cdot \alpha_{(1-\beta,s+s^*)}},$$

where we have assumed that $\frac{4L_{(\beta,s+s^*)}}{\alpha_{(1-\beta,s+s^*)}} < 9/10$, something that we shall establish below. Note that $\lambda_{\min}(\Sigma_0) > 0$ since $\Sigma$ is assumed to be invertible. In the random design settings we shall consider, we also have $\frac{\sqrt{L_{(\beta,s+s^*)}}}{\sqrt{\lambda_{\min}(\Sigma_0)} \cdot \alpha_{(1-\beta,s+s^*)}} = \mathcal{O}\left(\frac{1}{\sqrt{n}}\right)$. Then setting $\rho \leq \frac{1}{72}\epsilon^2 \cdot \lambda_{\min}(\Sigma_0) \cdot \alpha_{(1-\beta,s+s^*)}$ proves the convergence result.

As before, we can use the above result to establish sparse recovery guarantees in the statistical setting for Gaussian and sub-Gaussian design models. If our data matrix $X$ is generated from a Gaussian distribution $\mathcal{N}(\mathbf{0}, \Sigma)$ for some invertible $\Sigma$, then the results in Theorem 18 can be used to establish that $\Sigma^{-1/2}X$ satisfies the SRSC and SRSS properties at the required levels and that for $\alpha < \frac{1}{190}$ and $n \geq \Omega\left((2s + s^*)\log p + \log\frac{1}{\delta}\right)$, we have $\eta = \frac{2L_{(\beta,s+s^*)}}{\alpha_{(1-\beta,s+s^*)}} < 9/10$.

Thus, the above result can be applied with $\Sigma_0 = \Sigma$ to get convergence guarantees in the general Gaussian setting. We note that the above analysis can tolerate the same level of corruption as Theorem 4 and thus, we can improve the noise tolerance level to $\alpha \leq \frac{1}{65}$ here as well. We also note that these results can be readily extended to the sub-Gaussian setting as well. $\qquad\square$

## G  Robust Statistical Estimation

This section elaborates on how results on the convergence guarantees of our algorithms can be used to give guarantees for robust statistical estimation problems. We begin with a few definition of sampling models that would be used in our results.

**Definition 12.** *A random variable $x \in \mathbb{R}$ is called sub-Gaussian if the following quantity is finite*

$$\sup_{p\geq 1} p^{-1/2} \left(\mathbb{E}\,|x|^p\right)^{1/p}.$$

*Moreover, the smallest upper bound on this quantity is referred to as the sub-Gaussian norm of $x$ and denoted as $\|x\|_{\psi_2}$.*

**Definition 13.** *A vector-valued random variable $\mathbf{x} \in \mathbb{R}^p$ is called sub-Gaussian if its unidimensional marginals $\langle \mathbf{x}, \mathbf{v}\rangle$ are sub-Gaussian for all $\mathbf{v} \in S^{p-1}$. Moreover, its sub-Gaussian norm is defined as follows*

$$\|X\|_{\psi_2} := \sup_{\mathbf{v}\in S^{p-1}} \|\langle\mathbf{x},\mathbf{v}\rangle\|_{\psi_2}$$

We will begin with the analysis of Gaussian designs and then extend our analysis for the class of general sub-Gaussian designs.

**Lemma 14.** *Let $X \in \mathbb{R}^{p\times n}$ be a matrix whose columns are sampled i.i.d from a standard Gaussian distribution i.e. $\mathbf{x}_i \sim \mathcal{N}(\mathbf{0}, I)$. Then for any $\epsilon > 0$, with probability at least $1 - \delta$, $X$ satisfies*

$$s_{\max}(XX^\top) \leq n + (1-2\epsilon)^{-1}\sqrt{cnp + c'n\log\frac{2}{\delta}}$$

$$s_{\min}(XX^\top) \geq n - (1-2\epsilon)^{-1}\sqrt{cnp + c'n\log\frac{2}{\delta}},$$

*where $c = 24e^2\log\frac{3}{\epsilon}$ and $c' = 24e^2$.*

*Proof.* We will first use the fact that $X$ is sampled from a standard Gaussian to show that its covariance concentrates around identity. Thus, we first show that with high probability,

$$\left\| XX^\top - nI \right\|_2 \leq \epsilon_1$$

for some $\epsilon_1 < 1$. Doing so will automatically establish the following result

$$n - \epsilon_1 \leq s_{\min}(XX^\top) \leq s_{\max}(XX^\top) \leq n + \epsilon_1.$$

Let $A := XX^\top - I$. We will use the technique of covering numbers [18] to establish the above. Let $\mathcal{C}^{p-1}(\epsilon) \subset S^{p-1}$ be an $\epsilon$ cover for $S^{p-1}$ i.e. for all $\mathbf{u} \in S^{p-1}$, there exists at least one $\mathbf{v} \in \mathcal{C}^{p-1}$ such that $\|\mathbf{u} - \mathbf{v}\|_2 \leq \epsilon$. Standard constructions [18, see Lemma 5.2] guarantee such a cover of size at most $\left(1 + \frac{2}{\epsilon}\right)^p \leq \left(\frac{3}{\epsilon}\right)^p$. Now for any $\mathbf{u} \in S^{p-1}$ and $\mathbf{v} \in \mathcal{C}^{p-1}$ such that $\|\mathbf{u} - \mathbf{v}\|_2 \leq \epsilon$, we have

$$\left| \mathbf{u}^\top A\mathbf{u} - \mathbf{v}^\top A\mathbf{v} \right| \leq \left| \mathbf{u}^\top A(\mathbf{u} - \mathbf{v}) \right| + \left| \mathbf{v}^\top A(\mathbf{u} - \mathbf{v}) \right| \leq 2\epsilon \left\| A \right\|_2,$$

which gives us

$$\left\| XX^\top - nI \right\|_2 \leq (1 - 2\epsilon)^{-1} \cdot \sup_{\mathbf{v} \in \mathcal{C}^{p-1}(\epsilon)} \left| \left\| X^\top \mathbf{v} \right\|_2^2 - n \right|.$$

Now for a fixed $\mathbf{v} \in S^{n-1}$, the random variable $\left\| X^\top \mathbf{v} \right\|_2^2$ is distributed as a $\chi^2(n)$ distribution with $n$ degrees of freedom. Using Lemma 20, we get, for any $\mu < 1$,

$$\mathbb{P}\left[ \left| \left\| X^\top \mathbf{v} \right\|_2^2 - n \right| \geq \mu n \right] \leq 2\exp\left( -\min\left\{ \frac{\mu^2 n^2}{24ne^2}, \frac{\mu n}{4\sqrt{3}e} \right\} \right) \leq 2\exp\left( -\frac{\mu^2 n}{24e^2} \right).$$

Setting $\mu^2 = c \cdot \frac{p}{n} + c' \cdot \frac{\log \frac{2}{\delta}}{n}$, where $c = 24e^2 \log \frac{3}{\epsilon}$ and $c' = 24e^2$, and taking a union bound over all $\mathcal{C}^{p-1}(\epsilon)$, we get

$$\mathbb{P}\left[ \sup_{\mathbf{v} \in \mathcal{C}^{p-1}(\epsilon)} \left| \left\| X^\top \mathbf{v} \right\|_2^2 - n \right| \geq \sqrt{cnp + c'n \log \frac{2}{\delta}} \right] \leq 2\left( \frac{3}{\epsilon} \right)^p \exp\left( -\frac{\mu^2 n}{24e^2} \right) \leq \delta.$$

This implies that with probability at least $1 - \delta$,

$$\left\| XX^\top - nI \right\|_2 \leq (1 - 2\epsilon)^{-1} \sqrt{cnp + c'n \log \frac{2}{\delta}},$$

which gives us the claimed bounds on the singular values of $XX^\top$. □

**Theorem 15.** *Let $X \in \mathbb{R}^{p \times n}$ be a matrix whose columns are sampled i.i.d from a standard Gaussian distribution i.e. $\mathbf{x}_i \sim \mathcal{N}(\mathbf{0}, I)$. Then for any $\gamma > 0$, with probability at least $1 - \delta$, the matrix $X$ satisfies the SSC and SSS properties with constants*

$$\Lambda_\gamma^{Gauss} \leq \gamma n \left( 1 + 3e\sqrt{6\log \frac{e}{\gamma}} \right) + \mathcal{O}\left( \sqrt{np + n\log \frac{1}{\delta}} \right)$$

$$\lambda_\gamma^{Gauss} \geq n - (1 - \gamma)n \left( 1 + 3e\sqrt{6\log \frac{e}{1-\gamma}} \right) - \Omega\left( \sqrt{np + n\log \frac{1}{\delta}} \right).$$

*Proof.* For any fixed $S \in \mathcal{S}_\gamma$, Lemma 14 guarantees the following bound

$$s_{\max}(X_S X_S^\top) \leq \gamma n + (1 - 2\epsilon)^{-1} \sqrt{c\gamma np + c'\gamma n \log \frac{2}{\delta}}.$$

Taking a union bound over $\mathcal{S}_\gamma$ and noting that $\binom{n}{k} \leq \left( \frac{en}{k} \right)^k$ for all $1 \leq k \leq n$, gives us

$$\Lambda_\gamma \leq \gamma n + (1 - 2\epsilon)^{-1} \sqrt{c\gamma np + c'\gamma^2 n^2 \log \frac{e}{\gamma} + c'\gamma n \log \frac{2}{\delta}}$$

$$\leq \gamma n \left( 1 + (1 - 2\epsilon)^{-1} \sqrt{c' \log \frac{e}{\gamma}} \right) + (1 - 2\epsilon)^{-1} \sqrt{c\gamma np + c'\gamma n \log \frac{2}{\delta}},$$

which finishes the first bound after setting $\epsilon = 1/6$. For the second bound, we use the equality

$$X_S X_S^\top = X X^\top - X_{\bar{S}} X_{\bar{S}}^\top,$$

which provides the following bound for $\lambda_\gamma$

$$\lambda_\gamma \geq s_{\min}(X X^\top) - \sup_{T \in \mathcal{S}_{1-\gamma}} X_T X_T^\top = s_{\min}(X X^\top) - \Lambda_{1-\gamma}.$$

Using Lemma 14 to bound the first quantity and the first part of this theorem to bound the second quantity gives us, with probability at least $1 - \delta$,

$$\lambda_\gamma \geq n - \gamma' n \left( 1 + (1 - 2\epsilon)^{-1} \sqrt{c' \log \frac{e}{\gamma'}} \right) - (1 - 2\epsilon)^{-1} \left( 1 + \sqrt{\gamma'} \right) \sqrt{cnp + c'n \log \frac{2}{\delta}},$$

where $\gamma' = 1 - \gamma$. This proves the second bound after setting $\epsilon = 1/6$. $\qquad\square$

We now extend our analysis to the class of isotropic subGaussian distributions. We note that this analysis is without loss of generality since for non-isotropic sub-Gaussian distributions, we can simply use the fact that Theorem 3 can admit whitened data for calculation of the SSC and SSS constants as we did for the case of non-isotropic Gaussian distributions.

**Lemma 16.** *Let $X \in \mathbb{R}^{p \times n}$ be a matrix with columns sampled from some sub-Gaussian distribution with sub-Gaussian norm $K$ and covariance $\Sigma$. Then, for any $\delta > 0$, with probability at least $1 - \delta$, each of the following statements holds true:*

$$s_{\max}(X X^\top) \leq \lambda_{\max}(\Sigma) \cdot n + C_K \cdot \sqrt{pn} + t\sqrt{n}$$
$$s_{\min}(X X^\top) \geq \lambda_{\min}(\Sigma) \cdot n - C_K \cdot \sqrt{pn} - t\sqrt{n},$$

*where $t = \sqrt{\frac{1}{c_K} \log \frac{2}{\delta}}$, and $c_K, C_K$ are absolute constants that depend only on the sub-Gaussian norm $K$ of the distribution.*

*Proof.* Since the singular values of a matrix are unchanged upon transposition, we shall prove the above statements for $X^\top$. The benefit of this is that we get to work with a matrix with independent rows, so that standard results can be applied. The proof technique used in [18, Theorem 5.39] (see also Remark 5.40 (1) therein) can be used to establish the following result: with probability at least $1 - \delta$, with $t$ set as mentioned in the theorem statement, we have

$$\left\| \frac{1}{n} X X^\top - \Sigma \right\| \leq C_K \sqrt{\frac{p}{n}} + \frac{t}{\sqrt{n}}$$

This implies that for any $\mathbf{v} \in S^{p-1}$, we have

$$\left| \frac{1}{n} \| X^\top \mathbf{v} \|_2^2 - \mathbf{v}^\top \Sigma \mathbf{v} \right| = \left| \frac{1}{n} \mathbf{v}^\top X X^\top \mathbf{v} - \mathbf{v}^\top \Sigma \mathbf{v} \right| \leq \left| \frac{1}{n} X X^\top \mathbf{v} - \Sigma \mathbf{v} \right| \leq C_K \sqrt{\frac{p}{n}} + \frac{t}{\sqrt{n}}.$$

The results then follow from elementary manipulations and the fact that the singular values and eigenvalues of real symmetric matrices coincide. $\qquad\square$

**Theorem 17.** *Let $X \in \mathbb{R}^{p \times n}$ be a matrix with columns sampled from some sub-Gaussian distribution with sub-Gaussian norm $K$ and covariance $\Sigma$. Let $c_K, C_K$ and $t$ be fixed to values as required in Lemma 16. Note that $c_K$ and $C_K$ are absolute constants depend only on the sub-Gaussian norm $K$ of the distribution. Let $\gamma \in (0, 1]$ be some fixed constant. Then, with we have the following:*

$$\Lambda_\gamma^{subGauss(K,\Sigma)} \leq \left( \lambda_{\max}(\Sigma) \cdot \gamma + \sqrt{\frac{\gamma}{c_K} \log \frac{e}{\gamma}} \right) \cdot n + C_K \cdot \sqrt{\gamma pn} + t\sqrt{n}.$$

*Furthermore, fix any $\epsilon \in (0, 1)$ and let $\gamma$ be a value in $(0, 1)$ satisfying the following*

$$\gamma > 1 - \min \left\{ \frac{\epsilon \cdot \lambda_{\min}(\Sigma)}{\lambda_{\max}(\Sigma)}, \exp \left( 1 + W_{-1} \left( -\frac{c_K \epsilon^2 \cdot \lambda_{\min}^2(\Sigma)}{e} \right) \right) \right\},$$

*where $W_{-1}(\cdot)$ is the lower branch of the real valued restriction of the Lambert W function. Then we have, with the same confidence,*

$$\lambda_\gamma^{subGauss(K,\Sigma)} \geq (1 - 2\epsilon) \cdot \lambda_{\min}(\Sigma) \cdot n - C_K \left( 1 + \sqrt{1 - \gamma} \right) \sqrt{pn} - 2t\sqrt{n}$$

*Proof.* The first result follows from an application of Lemma 16, a union bound over sets in $\mathcal{S}_\gamma$, as well as the bound $\binom{n}{k} \leq \left(\frac{en}{k}\right)^k$ for all $1 \leq k \leq n$ which puts a bound on the number of sparse sets as $\log |\mathcal{S}_\gamma| \leq \gamma \cdot n \log \frac{e}{\gamma}$.

For the second result, we observe that $X_S X_S^\top = XX^\top - X_{\bar{S}} X_{\bar{S}}^\top$, so that $s_{\min}(X_S X_S^\top) \geq s_{\min}(XX^\top) - s_{\max}(X_{\bar{S}} X_{\bar{S}}^\top)$. This gives us

$$\inf_{S \in \mathcal{S}_\gamma} s_{\min}(X_S X_S^\top) \geq s_{\min}(XX^\top) - \sup_{S \in \mathcal{S}_{1-\gamma}} s_{\max}(X_S X_S^\top).$$

Using Lemma 16 and the first part of this result gives us

$$\inf_{S \in \mathcal{S}_\gamma} s_{\min}(X_S X_S^\top) \geq \lambda_{\min}(\Sigma) \cdot n - C_K \cdot \sqrt{pn} - t\sqrt{n}$$

$$- \left(\lambda_{\max}(\Sigma)(1-\gamma) + \sqrt{\frac{1-\gamma}{c_K} \log \frac{e}{1-\gamma}}\right) n - C_K \sqrt{(1-\gamma)pn} - t\sqrt{n}$$

$$= \left(\lambda_{\min}(\Sigma) - \lambda_{\max}(\Sigma)(1-\gamma) - \sqrt{\frac{1-\gamma}{c_K} \log \frac{e}{1-\gamma}}\right) n$$

$$- C_K \left(1 + \sqrt{1-\gamma}\right) \sqrt{pn} - 2t\sqrt{n}$$

$$\geq (1 - 2\epsilon) \cdot \lambda_{\min}(\Sigma) \cdot n - C_K \left(1 + \sqrt{1-\gamma}\right) \sqrt{pn} - 2t\sqrt{n},$$

where the last step follows from the assumptions on $\gamma$ and by noticing that it suffices to show the following two inequalities to establish the last step

1. $\lambda_{\max}(\Sigma)(1-\gamma) \leq \epsilon \cdot \lambda_{\min}(\Sigma)$
2. $(1-\gamma) \log \frac{e}{1-\gamma} \leq c_K \epsilon^2 \cdot \lambda_{\min}^2(\Sigma)$

The first part gives us the condition $\gamma > 1 - \frac{\epsilon \cdot \lambda_{\min}(\Sigma)}{\lambda_{\max}(\Sigma)}$ in a straightforward manner. For the second part, denote $v = c_K \epsilon^2 \cdot \lambda_{\min}^2(\Sigma)$. Note that for $v \geq 1$, *all* values of $\gamma \in (0,1]$ satisfy the inequality.

Otherwise we require the use of the Lambert W function (also known as the product logarithm function). This function ensures that its value $W(z)$ for any $z > -1/e$ satisfies $z = W(z)e^{W(z)}$. In our case, making a change of variable $(1-\gamma) = e^\eta$ gives us the inequality $(\eta - 1)e^{\eta-1} \geq -v/e$. Note that since $v \leq 1$ in this case, $-v/e \in (-1/e, 0)$ i.e. a valid value for the Lambert W function. However, $(-1/e, 0)$ is also the region in which the Lambert W function is multi-valued. Taking the worse bound for $\gamma$ by choosing the lower branch $W_{-1}(\cdot)$ gives us the second condition $\gamma \geq 1 - \exp\left(1 + W_{-1}\left(-\frac{c_K \epsilon^2 \cdot \lambda_{\min}^2(\Sigma)}{e}\right)\right)$. $\qquad \square$

It is important to note that for any $-1/e \leq z < 0$, we have $\exp\left(1 + W_{-1}(z)\right) > 0$ which means that the bounds imposed on $\gamma$ by Theorem 17 always allow a non-zero fraction of the data points to be corrupted in an adversarial manner. However, the exact value of that fraction depends, in a complicated manner, on the sub-Gaussian norm of the underlying distribution, as well as the condition number and the smallest eigenvalue of the second moment of the underlying distribution.

We also note that due to the generic nature of the previous analysis, which can handle the entire class of sub-Gaussian distributions, the bounds are not as explicitly stated in terms of universal constants as they are for the standard Gaussian design setting (Theorem 15).

We now establish that for a wide family of random designs, the SRSC and SRSS properties are satisfied with high probability as well. For sake of simplicity, we will present our analysis for the standard Gaussian design. However, the results would readily extend to general Gaussian and sub-Gaussian designs using techniques similar to Theorem 17.

**Theorem 18.** *Let $X \in \mathbb{R}^{p \times n}$ be a matrix whose columns are sampled i.i.d from a standard Gaussian distribution i.e. $\mathbf{x}_i \sim \mathcal{N}(\mathbf{0}, I)$. Then for any $\gamma > 0$ and $s \leq p$, with probability at least $1 - \delta$, the*

*matrix $X$ satisfies the SRSC and SRSS properties with constants*

$$L_{(\gamma,s)}^{Gauss} \leq \gamma n \left(1 + 3e\sqrt{6\log\frac{e}{\gamma}}\right) + \tilde{\mathcal{O}}\left(\sqrt{ns + n\log\frac{1}{\delta}}\right)$$

$$\alpha_{(\gamma,s)}^{Gauss} \geq n - (1-\gamma)n\left(1 + 3e\sqrt{6\log\frac{e}{1-\gamma}}\right) - \tilde{\Omega}\left(\sqrt{ns + n\log\frac{1}{\delta}}\right).$$

*Proof.* The proof of this theorem proceeds similarly to that of Theorem 15. Hence, we simply point out the main differences. First, we shall establish, that for any $\epsilon > 0$, with probability at least $1 - \delta$, $X$ satisfies the RSC and RSS properties at level $s$ with the following constants

$$L_s \leq n + (1-2\epsilon)^{-1}\sqrt{bns + b'n\log\frac{2}{\delta}}$$

$$\alpha_s \geq n - (1-2\epsilon)^{-1}\sqrt{bns + b'n\log\frac{2}{\delta}},$$

where $b = 24e^2\log\frac{3ep}{\epsilon s}$ and $b' = 24e^2$. To do so we notice that the only change needed to be made would be in the application of the covering number argument. Instead of applying the union bound over an $\epsilon$-cover $\mathcal{C}^{p-1}$ of $S^{p-1}$, we would only have to consider an $\epsilon$-cover $\mathcal{C}_s^{p-1}$ of the set $S_s^{p-1}$ of all $s$-sparse unit vectors in $p$-dimensions. A straightforward calculation shows us that

$$\left|\mathcal{C}_s^{p-1}\right| \leq \binom{p}{s}\left(1 + \frac{2}{\epsilon}\right)^s \leq \left(\frac{3ep}{\epsilon s}\right)^s.$$

Thus, setting $\mu^2 = b \cdot \frac{s}{n} + b' \cdot \frac{\log\frac{2}{\delta}}{n}$, where $b = 24e^2\log\frac{3ep}{\epsilon s}$ and $b' = 24e^2$, we get

$$\mathbb{P}\left[\sup_{\mathbf{v}\in\mathcal{C}_s^{p-1}}\left|\|X\mathbf{v}\|_2^2 - n\right| \geq \sqrt{bns + b'n\log\frac{2}{\delta}}\right] \leq \delta,$$

which establishes the required RSC and RSS constants for $X$. Now, moving on to the SRSS constant, it follows simply by applying a union bound over all sets in $\mathcal{S}_\gamma$ much like in Theorem 15. One can then proceed to bound the SRSC constant in a similar manner.

We note that the nature of the SRSC and SRSS bounds indicate that our TORRENT-FC algorithm in the high dimensional sparse recovery setting has noise tolerance properties, characterized by the largest corruption index $\alpha$ that can be tolerated, identical to its low dimnensional counterpart - something that Theorem 9 states explicitly. □

## H Supplementary Results

**Claim 19.** *Given any vector $\mathbf{v} \in \mathbb{R}^n$, let $\sigma \in S_n$ be defined as the permutation that orders elements of $\mathbf{v}$ in descending order of their magnitudes i.e. $\left|v_{\sigma(1)}\right| \geq \left|v_{\sigma(2)}\right| \geq \ldots \geq \left|v_{\sigma(n)}\right|$. For any $0 < p \leq q \leq 1$, let $S_1 \in \mathcal{S}_q$ be an arbitrary set of size $q \cdot n$ and $S_2 = \{\sigma(i) : n - p \cdot n + 1 \leq i \leq n\}$. Then we have $\|\mathbf{v}_{S_2}\|_2^2 \leq \frac{p}{q}\|\mathbf{v}_{S_1}\|_2^2 \leq \|\mathbf{v}_{S_1}\|_2^2$.*

*Proof.* Let $S_3 = \{\sigma(i) : n - q \cdot n + 1 \leq i \leq n\}$ and $S_4 = \{\sigma(i) : n - q \cdot n + 1 \leq i \leq n - p \cdot n\}$. Clearly, we have $\|\mathbf{v}_{S_3}\|_2^2 \leq \|\mathbf{v}_{S_1}\|_2^2$ since $S_3$ contains the smallest $q \cdot n$ elements (by magnitude). Now we have $\|\mathbf{v}_{S_3}\|_2^2 = \|\mathbf{v}_{S_2}\|_2^2 + \|\mathbf{v}_{S_4}\|_2^2$. Moreover, since each element of $S_4$ is larger in magnitude than every element of $S_2$, we have

$$\frac{1}{|S_4|}\|\mathbf{v}_{S_4}\|_2^2 \geq \frac{1}{|S_2|}\|\mathbf{v}_{S_2}\|_2^2.$$

This gives us

$$\|\mathbf{v}_{S_2}\|_2^2 = \|\mathbf{v}_{S_3}\|_2^2 - \|\mathbf{v}_{S_4}\|_2^2 \leq \|\mathbf{v}_{S_3}\|_2^2 - \frac{|S_4|}{|S_2|}\|\mathbf{v}_{S_2}\|_2^2,$$

which upon simple manipulations, gives us the claimed result. □

**Lemma 20.** *Let $Z$ be distributed according to the chi-squared distribution with $k$ degrees of freedom i.e. $Z \sim \chi^2(k)$. Then for all $t \geq 0$,*

$$\mathbb{P}\left[|Z - k| \geq t\right] \leq 2\exp\left(-\min\left\{\frac{t^2}{24ke^2}, \frac{t}{4\sqrt{3}e}\right\}\right)$$

*Proof.* This lemma requires a proof structure that traces several basic results in concentration inequalities for sub-exponential variables [18, Lemma 5.5, 5.15, Proposition 5.17]. The purpose of performing this exercise is to explicate the constants involved so that a crisp bound can be provided on the corruption index that our algorithm can tolerate in the standard Gaussian design case.

We first begin by establishing the sub-exponential norm of a chi-squared random variable with a single degree of freedom. Let $X \sim \chi^2(1)$. Then using standard results on the moments of the standard normal distribution gives us, for all $p \geq 2$,

$$(\mathbb{E}|X|^p)^{1/p} = ((2p-1)!!)^{1/p} = \left(\frac{(2p)!}{2^p p!}\right)^{1/p} \leq \frac{\sqrt{3}}{2}p$$

Thus, the sub-exponential norm of $X$ is upper bounded by $\sqrt{3}/2$. By applying the triangle inequality, we obtain, as a corollary, an upper bound on the sub-exponential norm of the centered random variable $Y = X - 1$ as $\|Y\|_{\psi_1} \leq 2\|X\|_{\psi_1} \leq \sqrt{3}$.

Now we bound the moment generating function of the random variable $Y$. Noting that $\mathbb{E}Y = 0$, we have, for any $|\lambda| \leq \frac{1}{2\sqrt{3}e}$,

$$\mathbb{E}\exp(\lambda Y) = 1 + \sum_{q=2}^{\infty} \frac{\mathbb{E}(\lambda Y)^q}{q!} \leq 1 + \sum_{q=2}^{\infty} \frac{(\sqrt{3}|\lambda|q)^q}{q!} \leq 1 + \sum_{q=2}^{\infty}(\sqrt{3}e|\lambda|)^q \leq 1 + 6e^2\lambda^2 \leq \exp(6e^2\lambda^2).$$

Note that the second step uses the sub-exponentially of $Y$, the third step uses the fact that $q! \geq (q/e)^q$, and the fourth step uses the bound on $|\lambda|$. Now let $X_1, X_2, \ldots X_k$ be $k$ independent random variables distributed as $\chi^2(1)$. Then we have $Z \sim \sum_{i=1}^{k} X_i$. Using the exponential Markov's inequality, and the independence of the random variables $X_i$ gives us

$$\mathbb{P}\left[Z - k \geq t\right] = \mathbb{P}\left[e^{\lambda(Z-k)} \geq e^{\lambda t}\right] \leq e^{-\lambda t}\mathbb{E}e^{\lambda(Z-k)} = e^{-\lambda t}\prod_{i=1}^{k}\mathbb{E}\exp(\lambda(X_i - 1)).$$

For any $|\lambda| \leq \frac{1}{2\sqrt{3}e}$, the above bounds on the moment generating function give us

$$\mathbb{P}\left[Z - k \geq t\right] \leq e^{-\lambda t}\prod_{i=1}^{k}\exp(6e^2\lambda^2) = \exp(-\lambda t + 6ke^2\lambda^2).$$

Choosing $\lambda = \min\left\{\frac{1}{2\sqrt{3}e}, \frac{t}{12ke^2}\right\}$, we get

$$\mathbb{P}\left[Z - k \geq t\right] \leq \exp\left(-\min\left\{\frac{t^2}{24ke^2}, \frac{t}{4\sqrt{3}e}\right\}\right).$$

Repeating this argument gives us the same bound for $\mathbb{P}\left[k - Z \geq t\right]$. This completes the proof. $\square$

# I  Supplementary Experimental Results

Figure 3: (a), (b), (c) Variation of recovery error with varying $p$, $\sigma$ and $n$. TORRENT was found to outperform DALM-$L_1$ in all these settings. (d) Recovery error as a function of runtime for various state-of-the-art $L_1$ solvers as indicated in [15].