[Reviews · NeurIPS 2015]

Submitted by Assigned_Reviewer_1

The idea of this paper is to use the hard thresholding to remove the potential outliers for robust regression. It seems a simple but interesting idea. One suggestion is that one might use a varying sequence of parameters for beta. For example, at the beginning stage, the estimation w^t is less accurate, and thus we might treat less samples as outliers; and as the algorithm goes with a more accurate w^t for large t, it might be OK to reject more outliers.

One difficulty with the proposed algorithm is how to choose the parameter beta and epsilon. More discussion needs to be added to the experiments.
Summary: The paper considers a robust regression algorithm by applying the hard thresholding to remove potential outliers.

Submitted by Assigned_Reviewer_2

Overall, this paper seemed to be well-written and clearly presented. However, the point about dealing with an additive noise perturbation in addition to sparse corruption is not quite clear. The main result here seems to be contained in Theorem 10 in Appendix A. But as Corollary 11 points out, the error bound provided by the theorem reduces to a constant plus epsilon in the case of Gaussian noise; i.e., the algorithm is not guaranteed to provide a consistent estimate of w^*. This seems to be a weaker conclusion than the results, say, in [5] or [6]. Does that mean the latter methods are preferable in the case of noisy observations? It would be good if the authors could clarify this point.
Summary: The results in this paper seem quite novel and relevant. The technical details also appear to be sound.

Submitted by Assigned_Reviewer_3

This paper proposes and analysis a new algorithm for the robust regression problem. Robust regression problem is an old problem, related to outliers detection.

Introduction ---------------

The robust regression problem is defined in Eq. (2). I think that the authors should slightly rephrase their definition as Eq. (2) is one possible "modern" definition of the robust least square problem.

Authors also claims that robust regression and sparse regression are "completely different". The only difference if the sparse assumption on the coefficients vector w. The robust regression has a much more specific model on the noise, but can also be a sparse regression (as studied in Sec. 5). Again, I think that the authors should rephrase this part to avoid miss-understanding.

My major remark here, is a lake of state of the art on robust regression/outliers detection. Indeed, only very recent work are mentioned, but this a very old problem, and some classical methods such as L1 regression (in the sense of min || y - X w ||_1 ) should be mentioned here, and used as a reference.

Sec 2 ------

The problem formulation is clear, and the noise is specified thanks to two terms : - a classic gaussian noise - an unbounded "sparse" noise.

However, this formulation is not the "only one" possible. Maybe one reference on some Bayesian model for the noise on robust regression would be welcome here, in order to have a quite complete point of view on robust regression.

Moreover, I think that the authors could reformulate clearly the robust least square regression problem as an optimisation problem such as (there is some others possibilities)

min_{w,b} 0.5 ||y - Xw - b||^2 + \lambda ||b||_0

Sec. 3 -------

This section introduces the "Torrent" algorithms studied in Sec. 4. This algorithms made me think of a kind of "matching pursuit": the selection of the active set is made directly on the residual instead on its correlation with the matrix X. Torrent-FC being the "OMP" and Torrent-GD the "classical" MP.

The definition of Hard thresholding suits well with the problem, but is a kind of counter-intuitive compared to the classical hard thresholding used for sparse regression.

Sec. 4 ------

This section presents the theoretical results of the proposed algorithm, and then is the main theoretical contribution of the paper. I have no particular remark on this part.

Sec. 5 -------

This section extends previous results to the "Robust sparse regression problem". Following my remark on Sec. 2,

I think that at this point, the author should re-formulate the problem as

min_{w,b} 0.5 ||y - Xw - b||^2 + \lambda ||b||_0 + \mu ||w||_0

Sec. 6 -------

The experimental part if ok, but the only method the authors use for comparison is L1-DALM which solve only the sparse robust regression problem, not the robust regression problem without the sparse assumption. So the comparison is not fair, as the goal of the two approached is not the same. I also think that the author must compared their algorithm to a classic L1-regression problem, which is the state of the art of robust regression (and also a L1-L1 approach for the robust sparse regression problem).

Summary: An OK paper on robust regression, with some theoretical results on the proposed algorithm, but with some lake on the presentation of robust regression as well as no comparison to some classical approached.

Submitted by Assigned_Reviewer_4

The presented work represents a significant contribution to the RLSR problem. The method, based on an alternate thresholding algorithm, is both clever and simple. The idea is to alternate, at each step,

the estimation of the set of "clean points", then to update the unknown parameter vector. The proposed original method (TORRENT-FC) guarantees exact recovery of the parameters vector, as long as the the proportion of outliers is below a fraction $\alpha$ (the corruption index) of the total data points, where $\alpha$, whose value is data dependent, is not larger than 1/2; and the subset strong convexity and subset strong smoothess properties (Definition 1) are satisfied; and n >= p log p, where p is the dimension of the data space; X is sampled from any sub-Gaussian distribution. However, the above method does not scale well to datasets with large p. This issue is overcome with a pair of modified versions, one based on gradient descent (TORRENT-GD) and the other a hybrid version of both algorithms, the so called TORRENT-HYBD. Moreover, both ensure geometric convergence. Experimental comparison shows that the proposed algorithms outperform the best L1-based algorithms: it is significantly faster and converges to a solution for which the estimation error for the parameters vector is much lower.

The TORRENT-HYB combines the best of TORRENT-FC and Torrent-GF. Indeed, while TORRENT-FC presents high improvement of the solution at each step but is computationally more expensive, while TORRENT-GV efficiently executes each step, but progress is slower.

Finally, the presented approach is extended to deal with high dimensional data, leading to the TORRENT-HD algorithm. This algorithm is able to yield exact recovery even for corruption indices as large as 0.7.

The article is well founded; the proofs of all theorems are present in the paper and in the supplementary material, and are very clear. It is also very well written paper, and as previously said, the proposed solution for this significant problem is very original.

Summary: This is a very interesting paper, where the authors propose a new Robust Least Squares Regression, that can handle up to 50% of outliers under mild conditions, less restrictive than previous methods: the data matrix is sampled from any sub-Gaussian distribution. The results achieved outperform those of state-of-the art L1 solvers, both in the estimation error and in convergence time.

Submitted by Assigned_Reviewer_5

This paper considers the problem of Robust Least Squares Regression where some of the responses of a linear model are corrupted by heavy noise or even adversarial manipulation and others are uncorrupted or corrupted only by iid Gaussian noise. So, the problem is a combination of the detection of the corrupted components and linear regression.

The authors propose to use iterative hard thresholding in the sense that in each iteration a least squares problem is solved w.r.t. to those data points for which the difference of response and linear prediction was smallest in the preceding step. Instead of solving the least squares problem exactly, a variant is proposed were the solution of the quadratic problem is approximated via a single gradient descent step. In addition a hybrid method is presented which chooses one of these options depending on how much the active set changed in the last iteration.

The idea of using iterative hard thresholding to find the "active set" of indices of the uncorrupted responses does not seem to be new, see, e.g., "Outlier detection using nonconvex penalized regression" by She & Owen (2011). However, the convergence results of Section 4 are new and interesting. It seems a bit unclear whether Theorems 3 and 4 implicitly assume that there exists a unique solution w^* or if there might be more than one and the algorithm finds one of them. Also, the constraint on n in Theorem 4 is given as an order of magnitude. However, the constant seem to matter since the problem probably gets easier the larger n is compared to p.

In general the paper is well written. Although mathematically correct, the phrasing in l. 90 that "\alpha < 1/2" for exact recovery is a bit misleading given that \alpha < 1/65 in Theorem 4.

Concerning the high-dimensional setting of Section 5, one should note that (3) cannot be solved globally optimally in general.

The L1 problem solved for comparison in Section 6 is min_w ||w||_1 + \lambda ||X'*w - y||_1. One could also use ||w||^2_2 in the low-dimensional case. Also, the experiments could be improved by applying the algorithms to real world data.

Minor issues: * Algorithm 2 should be called UPDATE_FC and Algorithm 3 UPDATE-GD

* In Section 2, the notation "bounded noise" is used for Gaussian noise and "potentially unbounded corruption" for the outliers. I am not sure if this is a good notation. There is no bound on the magnitude of Gaussian noise either.
Summary: The paper proposes iterative hard thresholding methods for Robust Least Squares Regression and provides conditions for exact recovery of the underlying model. Numerical experiments indicate advantages compared to L1-methods.

Author Feedback
Author rebuttal: We thank all the reviewers for their comments and suggestions. We will keep these in mind while revising the draft.

Reviewer 1
==========
- The modification to the algorithm proposed by the reviewer is interesting and one that we have actually tried. However we did not include it since we do not have a theoretical analysis of the resulting method. Moreover the method seems difficult to implement in practice since the rate of increasing beta seems to have an effect on convergence rates.

- Several interesting modifications are possible to the basic TORRENT framework. In this paper we focused on a class of simple algorithms with few tunable parameters that have solid theoretical analyses. It is noteworthy that TORRENT-FC has a single hyperparameter beta. Moreover, as Theorem 3 indicates, our bounds hold whenever beta is larger than the true fraction of outliers as long as there are enough clean responses.

Reviewer 2
==========
- Comparison to classical approaches like ||y - X^T w||_1: We do indeed cite and compare against such formulations. In fact, the formulation in line no. 67 can be easily re-written as
min_w ||w||_1 + \lambda ||y - X^T w||_1
by replacing b = y - X^T w. Note that for large values of lambda, this indeed minimizes ||y - X^T w||_1.

- Experiments - comparison to classical L1 and L1-L1 approaches: As mentioned above, the L1-DALM approach that we compare against can be reformulated as the "classical L1" approach mentioned by the reviewer. Hence, we do indeed compare against L1 and L1-L1 approaches that the reviewer mentions.

- L1-DALM only solves sparse regression: As mentioned above, L1-DALM solves ||w||_1 + \lambda ||y - X^T w||_1. This is equivalent to the low-d L1 problem ||y - X^Tw||_1 for large lambda. To ensure fairness, we tuned lambda over a fine grid (see line 368). Also, we compared against the L_2 regularized formulation (||w||_2^2+\lambda ||y-Xw||_1) but it also did not change the results significantly.

- Comparison to L1 solvers other than L1-DALM: as mentioned on line 373, we indeed compared against other state-of-the-art L1 solvers (see Fig 3 (d)).

- Sparse regression usually refers to problems where the regressor is sparse but the responses are not highly corrupted whereas robust regression refers to problems where response/covariates might be corrupted. Our comment was meant to highlight this fact.

- The noise model used in Section 2 is the most widely used model in literature that we could find. If the reviewer could point out other models then we would be happy to refer to them.

Reviewer 3
==========
We thank the reviewer for the kind comments.

Reviewer 4
==========
- Indeed our current analysis of the noise model with white noise and sparse corruptions does not guarantee a consistent estimate. In fact, due to adversarial corruptions, it seems unlikely that consistent estimates are even possible since the adversary can add corruptions only to responses where the Gaussian noise is "negative". Hence, the effective noise would not be zero-mean, which precludes consistent estimates.

- The result of [5] ([5, Theorem 4]) seems to be similar to ours as they also estimate the true w* up to the variance in the white noise. Moreover, the corruptions considered in [5] are not adaptive and need to be added independently of the data points and the white noise. The result of [6] does not even consider dense noise and even then, offers very weak noise tolerance (alpha < 1/sqrt(p)).

Reviewer 5
==========
We thank the reviewer for the kind comments. We will reformat the plots to increase their clarity.

Reviewer 6
==========
- We thank the reviewer for pointing out the reference of (She and Owen, 2011) and will appropriately reference the same.

- The results in our paper (Theorems 3 and 4) assume the noise model given in equation 2 (the more general model is given in Section 2) and guarantee convergence to the underlying parameter w* that generates the (corrupted) responses.

- We will explicate the constants in Theorem 4 in the supplementary material.

- We will state the actual bound (currently 1/65) in the introduction (line 90). We would like to stress that the main goal of the paper was to show that a constant fraction of corruptions in responses can be tolerated. We did not strive to optimize constants.

- As mentioned in line 368, we tune lambda on a fine grid. Hence, the "low-d" case i.e. one without ||w||_1 is already subsumed. Moreover, we did try using L_2 regularization, but it did not offer significant improvements because the data is sub-Gaussian and we provided enough clean responses for each experiment.

- Post submission, we also performed experiments on certain face recognition tasks that require robust regression problems to be solved. Here again we observe significantly better performance for our method than the state of the art for these tasks.